# UncertainGen: Uncertainty-Aware Representations of DNA Sequences for Metagenomic Binning

## Abstract

Metagenomic binning aims to cluster DNA fragments from mixed microbial samples into their respective genomes, a critical step for downstream analyses of microbial communities. Existing methods rely on deterministic representations, such as k-mer profiles or embeddings from large language models, which fail to capture the uncertainty inherent in DNA sequences arising from inter-species DNA sharing and from fragments with highly similar representations. We present the first probabilistic embedding approach, UncertainGen, for metagenomic binning, representing each DNA fragment as a probability distribution in latent space. Our approach naturally models sequence-level uncertainty, and we provide theoretical guarantees on embedding distinguishability. This probabilistic embedding framework expands the feasible latent space by introducing a data-adaptive metric, which in turn enables more flexible separation of bins/clusters. Experiments on real metagenomic datasets demonstrate the improvements over deterministic k-mer and LLM-based embeddings for the binning task by offering a scalable and lightweight solution for large-scale metagenomic analysis.

## 1 Introduction

Genomic sequences encode the blueprint of life, and analyzing them is fundamental for understanding biological processes, evolutionary relationships, and microbial ecosystems (Falkowski et al., 2008; Timmis et al., 2017; Cavicchioli et al., 2019). In recent years, advances in high-throughput DNA sequencing technologies have enabled large-scale studies of complex microbial communities directly from environmental samples. However, these technologies typically produce fragmented DNA sequences (called *reads*) rather than complete genomes. This fragmentation poses a significant challenge: recovering the full DNA sequences of the microbes in a sample requires assembling these reads and organizing them according to their origin.

The process of organizing reads from a mixed microbial sample is known as *metagenomic binning*, which aims to cluster DNA fragments so that each cluster corresponds to a distinct genome (Kunin et al., 2008). Accurate binning is critical for downstream analyses, such as functional annotation, phylogenetic profiling, and strain-level variation studies (Temperton & Giovannoni, 2012; Meyer et al., 2022). At its core, metagenomic binning relies on a representation of DNA fragments that preserves genomic similarity and inter-species dissimilarity, enabling meaningful comparisons between reads or assembled contiguous sequences (i.e. *contigs*).

Traditionally, DNA sequences are represented using *k-mer profiles*, wherein sequences are decomposed into substrings of length $k$ to construct the feature vectors of DNA fragments (see Figure 1 (a-c)). Numerous methods leverage these $k$-mer–based representations to learn latent representations (i.e., *embeddings*) to later cluster the DNA fragments for metagenomic binning (Teeling et al., 2004; Chan et al., 2008; Pan et al., 2023; Çelikkanat et al., 2024; Ji et al., 2021). Recent studies employs large language models that operate directly on raw sequences—eschewing explicit $k$-mer feature vectors—to generate embeddings with the aim of capturing richer contextual information (Nguyen et al., 2023; Zhou et al., 2023; 2024). However, recent works also indicate that $k$-mer–based embeddings achieve comparable performance while offering orders-of-magnitude greater computational efficiency than large genome foundation models (Çelikkanat et al., 2024).

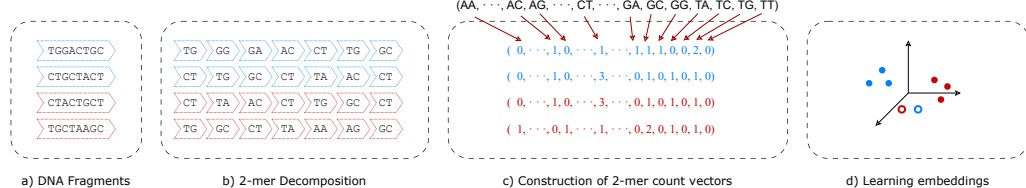

Figure 1: **Illustration of the metagenomic binning process**. Starting from a set of DNA sequences (a), the process ends with their two-dimensional embeddings derived from 2-mer profiles (d). In general, these embeddings allow the DNA fragments from two different species to be correctly clustered. However, the second and third DNA sequences in (a) pose an exception: although distinct, their $k$-mer representations shown in (c) are highly similar and, consequently, their embeddings are also very close (shown as the two empty circled points in (d)). Because the k-mer profiles of DNA within a species tend to be (locally) similar, the contrastive learning procedure attempts to position such fragments in both clusters but, since this is not possible, ultimately places them between them.

A shared characteristic of these methods is that they produce *deterministic embeddings*, mapping each DNA fragment to a single fixed point in the embedding space that is subsequently clustered and assigned to a single group representing the reconstructed species' DNA (Figure 1 illustrates this process). However, many DNA sequences can appear in multiple genomes—for instance through horizontal (lateral) gene transfer (Arnold et al., 2022)—and should ideally be assigned to their correct clusters. But this is impossible for any clustering algorithm because such sequences will be represented by the same point in the embedding space. A further limitation arises with $k$-mer–based representations: distinct sequences, potentially belonging to different clusters, can exhibit highly similar $k$-mer profiles and are therefore projected to similar embedding vectors, making it very difficult for the clustering algorithm to assign them accurately. Figure 1 visually illustrates this point.

In this work, we provide a mathematical formalization of the above issues and show that *deterministic embeddings* cannot resolve them. To address this limitation, we propose the use of *probabilistic embeddings* (Shi & Jain, 2019; Warburg et al., 2023; Karpukhin et al., 2024), where each fragment is mapped not to a single point but to a distribution (i.e., a region) in the embedding space. These distributions explicitly encode the ambiguity of fragments that may belong to multiple clusters and, more specifically, capture the uncertainty that a given $k$-mer profile can belong to multiple speies.

Previous work on probabilistic embeddings in computer vision, NLP, and graph representation learning have typically relied on heuristically selected distributional distances, such as the Kullback–Leibler or Wasserstein divergence to compare objects (Muzellec & Cuturi, 2018). In contrast, our framework employs a contrastive-learning formulation in which non-Euclidean distances between embeddings emerge naturally from a probability distribution defined over the embedding space. This formulation yields closed-form expressions for the expected pairwise likelihood, enabling efficient and scalable optimization. Moreover, we present a theoretical analysis identifying the types of sequences producing large covariance terms, thereby offering insight into how and why the model captures uncertainty arising from ambiguous or multi-class sequences.

To the best of our knowledge, we propose the first framework for probabilistic embeddings of DNA sequences for the metagenomic binning task that extends the $k$-mer-based representation approaches (Çelikkanat et al., 2024; Pan et al., 2023). We provide a scalable approach, UNCERTAINGEN, for embedding DNA sequences, offering both theoretical insights and practical performance gains.

- We introduce a novel probabilistic sequence embedding framework for metagenomic binning that models the uncertainty inherent in DNA sequences arising from inter-species DNA sharing and from fragments with highly similar representations

- We derive theoretical guarantees on the distinguishability of both deterministic and probabilistic embeddings, showing how the probabilistic embeddings expand the feasible latent space and improve the model's capacity to separate DNA fragments.

- We empirically demonstrate the effectiveness of our approach on real metagenomic datasets, showing improvements over deterministic k-mer and LLM-based embeddings.

The implementation of the proposed approach will be made publicly available after acceptance.

## 2 RELATED WORKS

**Embedding models for DNA sequences.** Representations of DNA sequences have advanced rapidly in recent years. Classical approaches are based on $k$-mer profiles (frequency vectors of length-$k$ substrings) (Wu et al., 2016; Kang et al., 2019), and many practical binners continue to rely on such features (Nissen et al., 2021; Wang et al., 2024; Kutuzova et al., 2024). More recently, the field has seen a surge of genome foundation models that adapt transformers or other long-context architectures to genomic data. DNABERT (Ji et al., 2021) introduced a BERT-style encoder with $k$-mer tokenization, while DNABERT-2 (Zhou et al., 2023) replaced fixed $k$-mers with byte-pair encoding (BPE) to improve efficiency and downstream performance. DNABERT-S (Zhou et al., 2024) further refined this line of work by introducing a curriculum contrastive learning strategy with manifold instance mixup loss to address the metagenomic binning task. HYENADNA (Nguyen et al., 2023) extended the context window further by modeling single-nucleotide tokens with a long-range convolutional architecture, reducing the cost of dense attention. Other approaches explore alternative geometries, such as HCNN Khan et al. (2025), which learns sequence representations in hyperbolic space.

In parallel, lightweight but task-specific models have been developed for metagenomics. SEMIBIN2 Pan et al. (2023) and related methods Wang et al. (2024) employ self-supervision and contrastive objectives tailored to binning, producing embeddings that cluster effectively by genome of origin. Recent work (Çelikkanat et al., 2024) has revisited the foundations of $k$-mer features, showing both their scalability and the limits of when $k$-mer profiles alone can separate genomes in practice. These results highlight a central trade-off: while foundation models offer expressive, context-aware representations, lightweight contrastive or $k$-mer-based approaches can rival or even outperform them in realistic binning scenarios.

**Probabilistic embeddings for contrastive learning.** Probabilistic embeddings have previously been explored, both generally (Warburg et al., 2023; Karpukhin et al., 2024; Bansal et al., 2025) and within a task specific context (Vilnis & McCallum, 2015; Shi & Jain, 2019). For example, in the context of face embeddings, Shi & Jain (2019) represents each images as a multivariate Gaussian distribution in embedding space, where a mutual likelihood score (MLS) is used to capture the likelihood of pairs of images belonging to the same person. Shi & Jain (2019) (Proposition 1) show that the proposed MLS score with fixed variance terms in the embedding space corresponds to a scaled and shifted negative squared Euclidean distance. Our results (Corollary 3.3.1) extends Proposition 1(Shi & Jain, 2019) by characterizing the (limited) expressivity of the equivalent of a fixed variance MLS score, while at the same time also showing that modeling capacity can be increased by allowing for varying covariance terms.

Probabilistic embedding have also been explored in a variational context. For instance, Oh et al. (2019) learns probabilistic embeddings using a soft contrastive loss while relying on a variational information bottleneck principle for optimization (Alemi et al., 2017). Jeong et al. (2025) reinterprets the InfoNCE loss as a reconstruction term in the ELBO objective through an approximation of the decoder model, which effectively also makes the representation decoder free. Kirchhof et al. (2023) posits a contrastive generative process and extends the InfoNCE loss to learn the correct posterior embedding distribution in latent space (up to rotation) for an unbounded number of negative samples; the correctness result relies on a known concentration parameter of the generative process for the positive samples. In contrast, we provide expressivity results related to model capacity, independent of any model specific parameters defining the data generating process.

## 3 PROPOSED MODEL

Let $\mathcal{S} \subset \Sigma^L$ be the set of sequences of length $L < \infty$ over alphabet $\Sigma := \{A, C, G, T\}$. In many genomic sequence clustering tasks, sequences originate from unknown genomes, and only sparse pairwise similarity information is available. In this regard, our goal is to learn an embedding function $\phi$ that captures the underlying cluster structure while also modeling the uncertainty in embedding space. Each cluster is expected to contain DNA fragments belonging to the same species.

**Objective:** We aim to learn an embedding function, $\phi$, that maps sequences into a latent space, where the distance reflects cluster membership. Specifically, for a given threshold $\tau > 0$, we require:

$$\|\phi(\mathbf{s}) - \phi(\mathbf{r})\| < \tau \quad \text{if and only if} \quad \ell(\mathbf{s}) = \ell(\mathbf{r}) = k \quad \text{for some } k \in [K], \tag{1}$$

where $\ell(\mathbf{s})$ denotes the cluster label of the DNA sequence $\mathbf{s} \in \mathcal{S}$.

Therefore, an embedding function satisfying this condition maps sequences from the same cluster close together, and sequences from different clusters remain well separated.

**Light-Weight Metagenomic Bining:** Our work builds on Çelikkanat et al. (2024); Pan et al. (2023), which introduced a state-of-the-art metagenomic binning algorithm. These methods achieve competitive accuracy while being several orders of magnitude faster than large genomic foundation models because they construct embeddings from efficient $k$-mer representations rather than using heavy sequence transformers. We adopt this principle of lightweight non-linear embeddings as the starting point for our approach.

In the works of Çelikkanat et al. (2024); Pan et al. (2023), each DNA sequence in the dataset is split into two equal-length segments to form a *positive* pair, while *negative* pairs are created by combining segments from two distinct sequences chosen at random. For every segment, we compute its $k$-mer profile and pass both profiles through a shared neural network that maps them into an embedding space. The contractive loss used there encourages embeddings of positive pairs to be close and embeddings of negative pairs to be far apart, thereby learning a genome-aware representation without supervision. These embeddings are later clustered with a standard algorithm. All DNA fragments in a cluster are assumed to belong to a single species.

Formally, let $\mathcal{S} = \{\mathbf{s}_i\}_{i=1}^N$ denote the set of DNA sequences in our dataset, with $\mathbf{s}_i^{(l)}$ and $\mathbf{s}_i^{(r)}$ being the left and right halves of each sequence. We construct triplets $\left\{(\mathbf{s}_i^{(l)}, \mathbf{s}_j^{(r)}, y_{ij})\right\}_{(i,j) \in \mathcal{I}}$, where $\mathcal{I}$ is the set of sequence index pairs, and $y_{ij} = 1$ if the two segments originate from the same sequence (positive) and $y_{ij} = 0$ otherwise (negative). The neural network parameters $\Omega$ are trained by minimizing

$$\mathcal{L}(\Omega) = -\sum_{(i,j) \in \mathcal{I}} \left[ y_{ij} \log P(Y_{ij} = 1 | \mathbf{s}_i^{(l)}, \mathbf{s}_j^{(r)}) + (1 - y_{ij}) \log\left(1 - P(Y_{ij} = 1 | \mathbf{s}_i^{(l)}, \mathbf{s}_j^{(r)})\right) \right], \tag{2}$$

with success probability $P(Y_{ij} = 1 | \mathbf{s}_i, \mathbf{s}_j) = \exp\left(-\|\phi_\Omega(\mathbf{s}_i) - \phi_\Omega(\mathbf{s}_j)\|^2\right)$, where $\phi_\Omega$ denotes the embedding function defined as a simple two-layer network with sigmoid activation functions.

Since we suppose that our dataset contains many different genomes, negative pairs are most likely to originate from different genomes. Similarly, the positive pairs contain DNA fragments from the same genome due to the nature of the construction procedure. Therefore, the model learns to produce similar embeddings for $k$-mer profiles from the same genome and dissimilar embeddings for profiles from different genomes.

**Motivation:** While the above contrastive framework provides an efficient way to learn genome-aware embeddings, its reliance on squared Euclidean distances between deterministic points (non-uncertain representations) in latent space imposes a fundamental limitation. As discussed in the introduction, many DNA fragments do not belong exclusively to a single cluster: they may genuinely occur in multiple genomes (e.g., through horizontal gene transfer (Arnold et al., 2022)) or, conversely, fragments from distinct genomes may yield indistinguishable $k$-mer feature vectors. In both cases, the fixed-point embeddings produced by the above model collapses such sequences to the same location in the latent space, preventing any clustering algorithm from assigning them consistently. Lemma 3.3 and Corollary 3.3.1 (below) formalize this limitation by showing that, under deterministic embeddings, it is not always possible to satisfy all pairwise constraints. In particular, when the set of DNA sequences is sufficiently large and originates from different clusters, they cannot simultaneously be mapped close to a cluster centroid to represent their membership while also being placed far apart from one another to reflect their pairwise dissimilarities.. This motivates our shift to probabilistic embeddings, where each fragment is represented by a distribution in the latent space, explicitly encoding ambiguity and thereby expanding the embedding space's flexibility to better separate multi-cluster or otherwise indistinguishable sequences.

**Proposed model:** Our approach uses two encoder networks outputting a mean–covariance pair $(\boldsymbol{\mu}, \mathbf{S})$ so that every fragment is represented as a Gaussian distribution. This provides a principled

way of modelling sequence-level ambiguity rather than evaluating the success probability $p(Y_{ij} = 1 \mid \mathbf{s}_i, \mathbf{s}_j)$ at fixed points. We define a new conditional distribution that *marginalizes over the uncertainty of both embeddings* $(\mathbf{z}_i, \mathbf{z}_j)$,

$$\mathbb{E}_{\substack{\mathbf{z}_i \sim \mathcal{N}(\mu_i, \mathbf{S}_i) \\ \mathbf{z}_j \sim \mathcal{N}(\mu_j, \mathbf{S}_j)}} [p(Y_{ij} = 1 \mid \mathbf{z}_i, \mathbf{z}_j)] = \mathbb{E}_{\substack{\mathbf{z}_i \sim \mathcal{N}(\mu_i, \mathbf{S}_i) \\ \mathbf{z}_j \sim \mathcal{N}(\mu_j, \mathbf{S}_j)}} \left[ \exp \left( -\frac{1}{2} (\mathbf{z}_i - \mathbf{z}_j)^\top \mathbf{K}_{ij}^{-1} (\mathbf{z}_i - \mathbf{z}_j) \right) \right], \quad (3)$$

where $\mathbf{K}_{ij} \succ 0$ is a positive definite matrix. The inner part of the expectation captures the (un-normalized) Gaussian likelihood of the embedding difference $\mathbf{z}_i - \mathbf{z}_j \sim \mathcal{N}(\mathbf{0}, \mathbf{K}_{ij})$, and $\mathbf{K}_{ij}$ can therefore be seen as representing the uncertainty in the distance between $\mathbf{z}_i$ and $\mathbf{z}_j$ providing different weights to the differences across each embedding dimension; $\mathbf{K}_{ij}$ can thus also be interpreted as a local metric tensor. By Lemma 3.1, we can find the closed-form solution of the expectation term over embeddings (proofs of all formal results are placed in Appendix A.2):

**Lemma 3.1.** *(Closed-form expectation) Let $\mathbf{z}_i \sim \mathcal{N}(\mu_i, \mathbf{S}_i)$ and $\mathbf{z}_j \sim \mathcal{N}(\mu_j, \mathbf{S}_j)$ be independent random variables. For a given positive definite matrix $\mathbf{K}_{ij} \succ 0$, Eq. 3 can be computed as*

$$\frac{1}{\sqrt{|\mathbf{K}_{ij}^{-1}(\mathbf{S}_i + \mathbf{S}_j) + \mathbf{I}|}} \exp \left( -\frac{1}{2} (\mu_i - \mu_j)^\top (\mathbf{S}_i + \mathbf{S}_j + \mathbf{K}_{ij})^{-1} (\mu_i - \mu_j) \right). \quad (4)$$

A natural choice for $\mathbf{K}_{ij}$ is $\mathbf{K}_{ij} := \alpha(\mathbf{S}_i + \mathbf{S}_j)$, so that (the local metric tensor) $\mathbf{K}_{ij}$ reflects the point-wise uncertainty in the embeddings with more "uncertain" points contributing less to the similarity measure. The parameter $\alpha$ is learnable, but in the remainder of the paper we set $\alpha = 1$; Appendix A.1 includes further insight into the role of $\alpha \in \mathbb{R}^+$. With this choice, the expectation in Eq. 3 simplifies algebraically to

$$\mathbb{E}_{\substack{\mathbf{z}_i \sim \mathcal{N}(\mu_i, \mathbf{S}_i) \\ \mathbf{z}_j \sim \mathcal{N}(\mu_j, \mathbf{S}_j)}} [p(Y_{ij} = 1 \mid \cdots)] = \frac{1}{\sqrt{2^D}} \exp \left( -\frac{1}{4} (\mu_i - \mu_j)^\top (\mathbf{S}_i + \mathbf{S}_j)^{-1} (\mu_i - \mu_j) \right), \quad (5)$$

where $D$ is the latent dimension size. Here, it is worth emphasizing that both the mean vectors $(\mu_i, \mu_j)$ and covariance matrices $(\mathbf{S}_i, \mathbf{S}_j)$ are parameterized by two simple neural networks denoted by $\phi_\mu$ and $\phi_\sigma$. In our experimental setup, they consist of a single hidden layer including 512 units with sigmoid activation functions, and the output dimension (i.e., $D$) is set to 256.

It is important to note that the expectation in Eq. 5 does not yield a properly normalized Bernoulli success probability, because its value ranges only from 0 up to $1/\sqrt{2^D}$ rather than the full $[0, 1]$ interval. To obtain a valid probability measure, we therefore renormalize this quantity by multiplying it with $\sqrt{2^D}$. This rescaling defines our final success probability, denoted by $q(Y_{ij} = 1 \mid s_i, s_j)$, which is guaranteed to lie between 0 and 1.

We optimize the parameters of these neural networks by maximizing the same loss given in Eq. 2, but using $q(Y_{ij} = 1 \mid s_i, s_j)$ as a sucess probability for positive pairs.

**Definition 3.2.** *For a given $\epsilon \in (0, 1/2)$, a mapping function $\phi : \mathcal{S} \to \mathbb{R}^D \times \mathbb{R}_+^D$, where $\phi := (\phi_\mu, \phi_\sigma)$ with $\phi_\mu : \mathcal{S} \to \mathbb{R}^D$ and $\phi_\sigma : \mathcal{S} \to \mathbb{R}_+^D$, satisfying*

$$q(Y_{ij} = y_{ij} \mid s_i, s_j) \geq (1 - \epsilon)$$

*for all $((\mathbf{s}_i, \mathbf{s}_j), y_{ij}) \in \mathcal{S} \times \mathcal{Y}$ is called an $\epsilon$-**distinguishable embedding** function for $\mathcal{S} \times \mathcal{Y}$, where $\mathcal{S} \subset \Sigma^L \times \Sigma^L$ denote the set of sequence pairs with associated labels $\mathcal{Y}$.*

For notational convenience, we omit $\mathcal{Y}$ whenever it is clear from the context. Intuitively, an $\epsilon$-distinguishable embedding function guarantees that positive pairs remain close in the latent space, while negative pairs are sufficiently separated. Lemma 3.3 formalizes this relationship by providing explicit bounds on the Euclidean distance between the embedding means as a function of the corresponding variances, thus offering theoretical guarantees for pairwise distinguishability.

**Lemma 3.3.** *Let $\epsilon \in (0, 1/2)$, and let $\phi : \mathcal{S} \to \mathbb{R}^D \times \mathbb{R}_+^D$ be an $\epsilon$-distinguishable embedding function for a pair $(\mathbf{s}_i, \mathbf{s}_j) \in \mathcal{S}^2$ and label $y_{ij} \in \{0, 1\}$ where $\phi := (\phi_\mu, \phi_\sigma)$ with $\phi_\mu : \mathcal{S} \to \mathbb{R}^D$ and $\phi_\sigma : \mathcal{S} \to \mathbb{R}_+^D$. Then the following bounds hold:*

$$\min_d \{(\phi_\sigma(\mathbf{s}_i) + \phi_\sigma(\mathbf{s}_j))_d\} \log \left( \frac{1}{\epsilon^4} \right) \leq \|\phi_\mu(\mathbf{s}_i) - \phi_\mu(\mathbf{s}_j)\|_2^2 \qquad \text{if } y_{ij} = 0, \quad (6)$$

$$\max_d \{(\phi_\sigma(\mathbf{s}_i) + \phi_\sigma(\mathbf{s}_j))_d\} \log \left( \frac{1}{(1-\epsilon)^4} \right) \geq \|\phi_\mu(\mathbf{s}_i) - \phi_\mu(\mathbf{s}_j)\|_2^2 \qquad \text{if } y_{ij} = 1. \quad (7)$$

These bounds highlight the critical role of variances in our setting. Intuitively, if a sequence is associated with multiple clusters, its mean representation cannot be simultaneously close to all corresponding centroids, since these centroids must remain sufficiently separated. For instance, Figure 2 depicts six blue points placed at distance $\tau$ from a central green point, while also being pairwise $\tau$-separated. In this two-dimensional configuration, it is not possible to place an additional point (such as the red point) that lies within distance $\tau$ of the green point while remaining more than $\tau$ away from all the blue points. Hence, the covariance terms $\phi_\sigma(\mathbf{s})$ must adaptively increase to satisfy the distinguishability condition in Eq. 3.2.

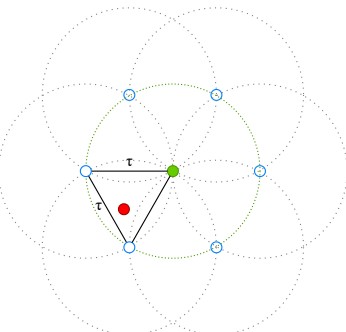

Figure 2: Packing number $(\mathcal{P}_\tau^D)$

Before stating the following corollary, we need to first introduce the *packing number* $\mathcal{P}_\tau^D$ which is the maximum number of $\tau$-separated distinct points in a ball of radius $\tau$ in $D$-dimensional space (Vershynin, 2018). It will help us to formalize the fundamental limitations of the embedding function.

**Corollary 3.3.1.** *Let $\phi : \mathcal{S} \to \mathbb{R}^D \times \mathbb{R}_+^D$ be an embedding function for the set $\mathcal{S}$ where $\phi := (\phi_\mu, \phi_\sigma)$ with $\phi_\mu : \mathcal{S} \to \mathbb{R}^D$ and $\phi_\sigma : \mathcal{S} \to \mathbb{R}_+^D$. If $\phi_\sigma(\mathbf{s}_i)_d = \phi_\sigma(\mathbf{s}_j)_d$ for all $\mathbf{s}_i, \mathbf{s}_j \in \mathcal{S}$, and $\forall d \in [D]$, and if there exists $\mathcal{P}_\tau^D + 2$ sequences, $\mathbf{s}_0, \mathbf{s}_1, \ldots, \mathbf{s}_{\mathcal{P}_\tau^D+1} \in \mathcal{S}$ such that each $(\mathbf{s}_0, \mathbf{s}_i)$ is a positive pair (i.e. $y_{(0,i)} = 1$) for all $i \in \{1, \ldots \mathcal{P}_\tau^D + 1\}$ and $(\mathbf{s}_i, \mathbf{s}_j)$ is a negative pair (i.e. $y_{(i,j)} = 0$) for $1 \le i < j \le \mathcal{P}_D + 1$, then it cannot be $\epsilon$-distinguishable function for $\epsilon \in (0, 1/2)$.*

From Corollary 3.3.1, we see that fixed-variance embeddings have intrinsic limitations in expressiveness: they cannot simultaneously satisfy the pairwise constraints of a sufficiently large sequence set. Allowing covariance terms to vary introduces additional degrees of freedom, enhancing the modeling capacity of the embedding function. Theorem 3.4 relies on this insight, showing that sequences belonging to multiple clusters tend to have larger covariance terms in order to handle the desired complex proximity relationship among sequences, building on distances in a latent space.

**Theorem 3.4.** *An embedding function $\phi : \mathcal{S} \to \mathbb{R}^D \times \mathbb{R}_+^D$ with bounded means (i.e. $\|\phi_\mu(\mathbf{s})\| < \infty$) is $\epsilon$-distinguishable for some $\epsilon \in (0, 1/2)$ if and only if there exists a set of sequences $\{\mathbf{s}_0, \mathbf{s}_1, \ldots, \mathbf{s}_N\} \subseteq \mathcal{S}$ where each $(\mathbf{s}_0, \mathbf{s}_i)$ is a positive pair and $(\mathbf{s}_i, \mathbf{s}_j)$ is negative pair satisfying $\phi_\sigma(\mathbf{s}_i)_d < \infty$ for $1 \le i \le N$ and $\phi_\sigma(\mathbf{s}_0)_d \to \infty$ for all $d \in [D]$ with $N > P_\tau^D$.*

This result underscores the importance of probabilistic embeddings: by having covariance terms, the model can represent complex relationships among sequences that deterministic embeddings cannot capture. In the following section, we will demonstrate the effectiveness of this approach on artificial and real genomic datasets.

# 4 EXPERIMENTS

We evaluated UNCERTAINGEN under the same experimental setup as (Zhou et al., 2024; Çelikkanat et al., 2024) to ensure a fair comparison with previous deterministic and large genome foundation models while highlighting the benefits of probabilistic embeddings for metagenomic binning. Due to page limitations, we provide the detailed information about the baseline models in Appendix B.

**Datasets.** For our experiments, we adopt the benchmark datasets introduced in prior work on the metagenomic binning task (Zhou et al., 2024). The datasets are constructed from reference genomes in GenBank and consist of viral, fungal, and bacterial sequences. The training data contains more than 2 million sequence pairs of length 1000bp. For testing, we have six datasets (*Reference 5/6*, *Plant 5/6*, and *Marine 5/6*) with species represented by highly variable numbers of sequences ($10$–$4{,}599$), ranging from 2-20 kbp in length. While *Reference* datasets consist of DNA fragments from 250-330 fungal and viral genomes, and *Marine* and *Plant*-associated environments contain 70k-125k sequences from roughly 180-520 species.

**Training procedure.** For our method, training was performed within a contrastive learning framework by optimizing the objective function given in Eq. 2 with our new success probability

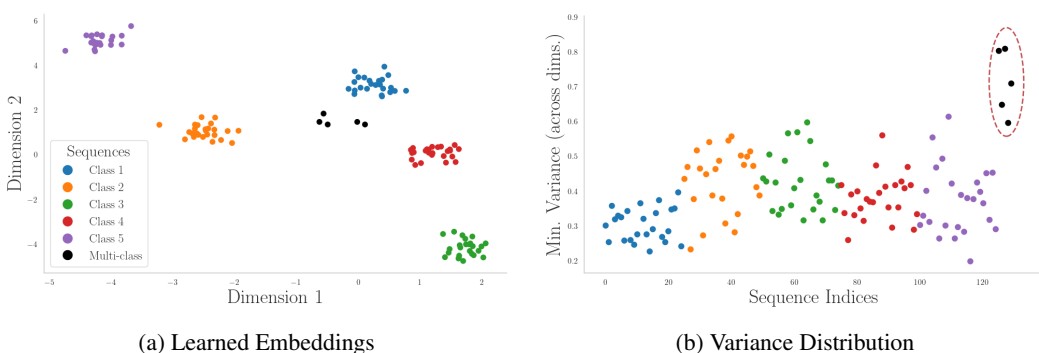

(a) Learned Embeddings

(b) Variance Distribution

Figure 3: Visualization of the learned embeddings and the variance distribution of the sequences.

$q(y_{ij} \mid \cdots)$. Positive pairs were obtained by splitting each fragment into two halves, guaranteeing that both subsequences come from the same genome. Negative pairs were formed by randomly pairing fragments from the dataset, and this procedure ensures, with high probability, that the paired sequences come from different genomes.

We trained our model using the Adam optimizer with a learning rate of $10^{-2}$. The model consists of 2 two-layer neural networks, as described in Section 3, that output the mean and variance terms of the multivariate normal distribution for a given input sequence. To improve stability, we first train the mean network alone for 50 epochs, and then train the variance network only for an additional 20 epochs. From the original dataset consisting of $2 \times 10^6$ pairs (Zhou et al., 2024), we randomly subsample $10^5$ pairs to demonstrate that our method is effective even with smaller training sets compared to large genome foundation models. For each positive pair, we generated 200 negative pairs, resulting in a total of $2,01 \times 10^7$ pairs. Training was also performed with a batch size of $10^5$.

### 4.1 TOY EXAMPLE WITH $k$-MER DATASET

To investigate the behavior of our model in a controlled setting, we designed a synthetic dataset of 4-mer sequences. This setup allows us to assess whether the model can learn meaningful low-dimensional embeddings that reflect cluster structure and how it represents sequences that span multiple classes. Due to the space limitations, we provide the details in Appendix B.1.

We generated sequences of length 100 from multinomial distributions defining 5 distinct classes, each with a characteristic 4-mer compositions. To simulate ambiguity, we additionally generated 5 "multi-class" sequences by combining $k$-mer counts from multiple classes, representing inputs that do not belong exclusively to a single class. This design allows us to evaluate how the model handles both well-separated clusters and overlapping class memberships.

Positive sequence pairs were formed by sampling sequences from the same class, while negative pairs were drawn across different classes, which introduces the possibility of false negatives. For multi-class sequences, pairs were also constructed with sequences from their contributing classes, allowing the model to learn representations that account for both pure and mixed memberships.

We learn the sequence embeddings in a 2-dimensional latent space, enabling direct visualization of the learned geometry without relying on dimensionality reduction techniques that could distort structural relationships (Figure 3a). The resulting embeddings reveal well-separated clusters corresponding to distinct species, while sequences that belong to multiple classes occupy intermediate regions (black points). For each sequence, our model also predicts a diagonal covariance matrix, which quantifies the degree of uncertainty in its placement. This uncertainty is particularly shown for sequences belonging to multiple classes, as reflected in their larger covariance values in Figure 3b. In line with our theoretical results (Lemma 3.3 and Theorem 3.4), the minimum variance across dimensions (i.e. $\min_d\{(\phi_\sigma(\mathbf{s}))_d\}$) provides a lower bound on pairwise distances, and sequences associated with multiple classes indeed display a higher minimum variance. This confirms that the model not only separates clusters effectively, but also encodes the uncertainty of ambiguous cases.

Table 1: Detailed comparison of REVISITKMERS and UNCERTAINGEN. The counts indicate the number of detected high-quality bins (i.e., number of clusters whose $F_1$-score is greater than 0.9).

|  | *Reference 5* | *Plant 5* | *Marine 5* | *Reference 6* | *Plant 6* | *Marine 6* |
|---|---|---|---|---|---|---|
| REVISITKMERS | 126 | 29 | 112 | 128 | 28 | 125 |
| UNCERTAINGEN | 135 | 32 | 124 | 132 | 23 | 127 |

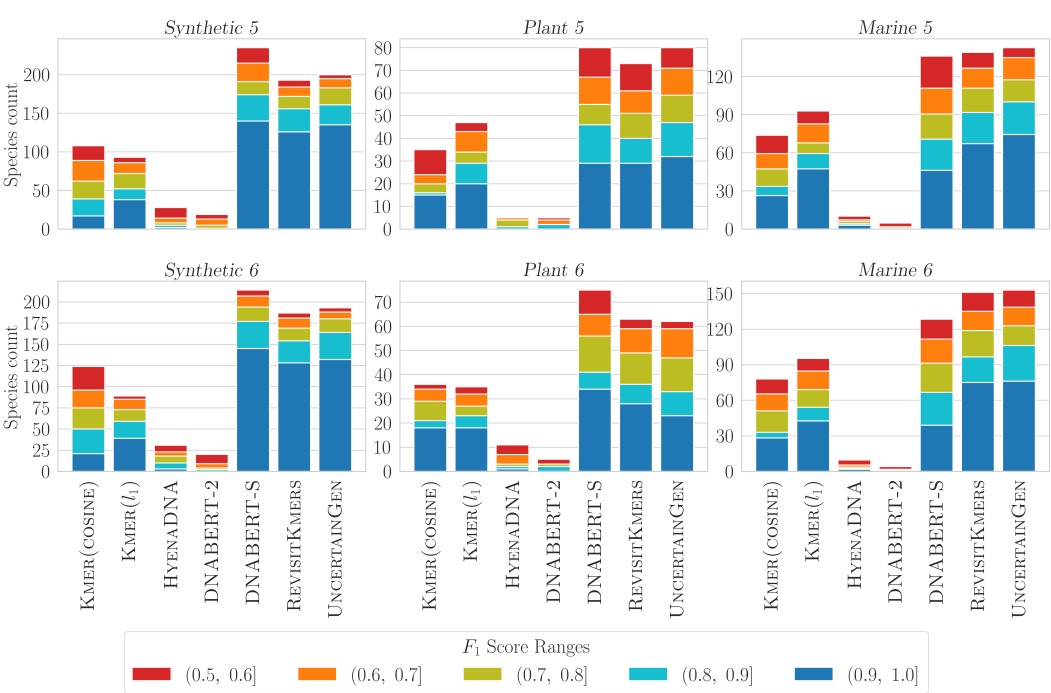

Figure 4: Metagenomic binning results. Cluster counts are segmented by F1-score quality ranges. The dark blue portion highlights the highest-quality bins for each model-dataset combination.

## 4.2 METAGENOMICS BINNING

We evaluate our methods on the metagenomic binning task, where the objective is to cluster sequences into species-level groups without prior knowledge of the number of clusters. In this regard, we adopt the modified K-Medoid algorithm of Zhou et al. (2024), which jointly estimates the cluster assignments and the underlying number of species. This setting is particularly challenging as it requires models to provide representations that are simultaneously discriminative and robust under unsupervised partitioning. For computing similarities between sequences, we employ cosine similarity for all genome-scale foundation models as well as the KMERS(COSINE) baseline. In contrast, we use an exponential kernel over the $\ell_1$ distance for KMERS($\ell_1$), and $\ell_2$ distance for REVITK-MERS. We use an exponential kernel over the generalized Mahalanobis term in Eq. 5 as a natural choice for our model.

Following established evaluation strategies, we stratify clusters into 5 quality tiers based on their $F_1$ scores. High-quality bins, defined as clusters with $F_1 > 0.9$, are highlighted in dark blue in Figure 4. Across datasets, UNCERTAINGEN consistently outperforms its deterministic counterpart, with the sole exception of the *Plant-6* dataset (see Table 1 for a detailed comparison). Moreover, while the strongest competing genome-scale foundation model, DNABERT-S, achieves slightly higher performance on the *Reference* dataset, our method surpasses it on the *Marine* dataset when focusing on high-quality bins. These results demonstrate that our approach is not only competitive with state-of-the-art foundation models but also offers the added benefit of a principled probabilistic formulation, enabling more robust and interpretable clustering in metagenomic settings with a smaller number of parameters.

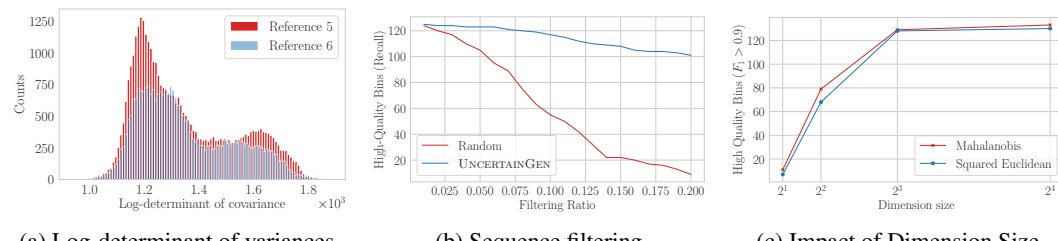

| (a) Log-determinant of variances. | (b) Sequence filtering | (c) Impact of Dimension Size |

Figure 5: Ablations studies examining the behavior of the proposed *UncertainGen* model.

## 4.3 ABLATION STUDIES

To better understand the behavior of our model, we perform a series of ablation studies. Due to space constraints, we discuss the results for the *Reference* datasets in the main text but a more comprehensive analysis across all datasets is provided in Appendix B.2.

**Distribution of variances.** We first analyze the distribution of the predicted variance terms for sequences in the test data. Specifically, we report $\sum_{d=1} \log\left(\phi_\sigma(\mathbf{s})_d + 1\right)$, which aggregates uncertainty across dimensions. As shown in Figure 5a, both variants of the *Reference* dataset exhibit multimodal distributions, suggesting that the model captures non-trivial heterogeneity in sequence-level uncertainty. This indicates that the variance estimates are not merely noise but encode meaningful structure about the underlying sequence distributions.

**Sequence filtering.** We filter out sequences with the largest log-determinant values to examine which sequences the model identifies as uncertain, and we report the number of clusters having a recall score $\geq 0.9$ (Figure 5b). For comparison, we also include a random filtering baseline. To ensure fairness, removed sequences are assigned to a "garbage" label so that they do not artificially inflate false negatives. Our results show that filtering by model uncertainty consistently retains a larger number of high-recall clusters compared to random filtering. This shows that the model assigns higher uncertainty to sequences from low-quality bins or to those that would otherwise contribute to false negatives, thereby acting as an effective mechanism for uncertainty-aware sequence selection.

**Dimension size.** As discussed in our theoretical analysis (Section 3), incorporating covariance terms provides the model with additional representational capacity compared to squared Euclidean distance: beyond capturing predictive uncertainty, the embedding space approximates a non-Euclidean Riemannian manifold. This enhanced geometry allows the model to better separate complex sequence structures. Empirical results in Figure 5c support this intuition. We observe consistent improvements when covariance terms are included, with the gains being especially pronounced in lower-dimensional settings where representational bottlenecks are most restrictive.

## 5 CONCLUSIONS, LIMITATIONS AND FUTURE WORKS

We introduced UNCERTAINGEN, the first probabilistic embedding framework for metagenomic binning. Unlike deterministic k-mer or LLM-based representations, our method maps each DNA fragment to a distribution in latent space, explicitly encoding sequence-level uncertainty. Theoretical analysis showed how variance terms enlarge the feasible embedding space and improve pairwise distinguishability, while experiments on both synthetic and real metagenomic data demonstrated consistent gains in binning quality over strong deterministic baselines.

Our study also reveals some limitations. We relied on a simplified semi-supervised pairing strategy and two-layer networks; more expressive architectures or richer positive/negative sampling schemes may further enhance performance. In addition, although variance terms estimates uncertainty, whether these estimates are calibrated or not remains an open question.

Future work will extend UNCERTAINGEN beyond k-mer based representations, explore hierarchical or non-Gaussian distributions for even richer uncertainty modeling, and integrate our embeddings into end-to-end pipelines for metagenome reconstruction. We hope this framework stimulates broader adoption of uncertainty-aware representations in computational genomics.

## Reproducibility

We will make our code publicly available upon acceptance and provide an accessible link in the paper. In the supplementary materials, we include the datasets and an anonymized version of the implementation to enable verification during the review process. The hyperparameter configurations and training procedures used in our experiments are described in Appendix B. In addition, all theoretical contributions are supported by complete proofs, which are also provided in Appendix A.2. Together, these resources ensure that our results can be reliably reproduced.

## Use of Large Language Models (LLMs)

LLMs were used to enhance the clarity and readability of the paper. Their use was limited to improving the writing style and rephrasing certain statements.

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

## A  APPENDIX

### A.1   THE ROLE OF $\alpha$ IN $\mathbf{K}_{ij} = \alpha(\mathbf{S}_i + \mathbf{S}_j)$

Consider the closed form expectation of Lemma 3.1:

$$\mathbb{E}_{\substack{\mathbf{z}_i \sim \mathcal{N}(\mu_i, \mathbf{S}_i) \\ \mathbf{z}_j \sim \mathcal{N}(\mu_j, \mathbf{S}_j)}} \left[ \exp\left( -\frac{1}{2}(\mathbf{z}_i - \mathbf{z}_j)^\top \mathbf{K}_{ij}^{-1}(\mathbf{z}_i - \mathbf{z}_j) \right) \right]$$

$$= \frac{1}{\sqrt{|\mathbf{K}_{ij}^{-1}(\mathbf{S}_i + \mathbf{S}_j) + \mathbf{I}|}} \exp\left( -\frac{1}{2}(\mu_i - \mu_j)^\top (\mathbf{S}_i + \mathbf{S}_j + \mathbf{K}_{ij})^{-1} (\mu_i - \mu_j) \right). \quad (8)$$

By setting $\mathbf{K}_{ij} = \alpha(\mathbf{S}_i + \mathbf{S}_j)$ we have

$$\frac{1}{\sqrt{|\mathbf{K}_{ij}^{-1}(\mathbf{S}_i + \mathbf{S}_j) + \mathbf{I}|}} = \frac{1}{\sqrt{|\alpha^{-1}\mathbf{I} + \mathbf{I}|}} = \frac{1}{\sqrt{|(1 + \alpha^{-1}\mathbf{I}|}}$$

$$= (1 + \alpha^{-1})^{-D/2}$$

and

$$-\frac{1}{2}(\mu_i - \mu_j)^\top (\mathbf{S}_i + \mathbf{S}_j + \mathbf{K}_{ij})^{-1} (\mu_i - \mu_j)$$

$$= -\frac{1}{2}(\mu_i - \mu_j)^\top (\mathbf{S}_i + \mathbf{S}_j + \alpha(\mathbf{S}_i + \mathbf{S}_j))^{-1} (\mu_i - \mu_j)$$

$$= -\frac{1}{2}(\mu_i - \mu_j)^\top ((1 + \alpha)(\mathbf{S}_i + \mathbf{S}_j))^{-1} (\mu_i - \mu_j)$$

$$= -\frac{1}{2(1 + \alpha)}(\mu_i - \mu_j)^\top (\mathbf{S}_i + \mathbf{S}_j)^{-1} (\mu_i - \mu_j).$$

Hence

$$\frac{1}{\sqrt{|\mathbf{K}_{ij}^{-1}(\mathbf{S}_i + \mathbf{S}_j) + \mathbf{I}|}} \exp\left( -\frac{1}{2}(\mu_i - \mu_j)^\top (\mathbf{S}_i + \mathbf{S}_j + \mathbf{K}_{ij})^{-1} (\mu_i - \mu_j) \right)$$

$$= (1 + \alpha^{-1})^{-D/2} \exp\left( -\frac{1}{2(1 + \alpha)}(\mu_i - \mu_j)^\top (\mathbf{S}_i + \mathbf{S}_j)^{-1} (\mu_i - \mu_j) \right)$$

$$\to \begin{cases} 1 & \text{as } \alpha \to \infty \\ 0 & \text{as } \alpha \to 0 \end{cases}.$$

### A.2   THEORETICAL ANALYSIS

**Lemma A.1.** *(Closed-form expectation) Let $\mathbf{z}_i \sim \mathcal{N}(\mu_i, \mathbf{S}_i)$ and $\mathbf{z}_j \sim \mathcal{N}(\mu_j, \mathbf{S}_j)$ be independent random variables. For a given positive definite matrix $\mathbf{K}_{ij} \succ 0$, the expectation term is equal to*

$$\mathbb{E}_{\substack{\mathbf{z}_i \sim \mathcal{N}(\mu_i, \mathbf{S}_i) \\ \mathbf{z}_j \sim \mathcal{N}(\mu_j, \mathbf{S}_j)}} \left[ \exp\left( -\frac{1}{2}(\mathbf{z}_i - \mathbf{z}_j)^\top \mathbf{K}_{ij}^{-1}(\mathbf{z}_i - \mathbf{z}_j) \right) \right]$$

$$= \frac{1}{\sqrt{|\mathbf{K}_{ij}^{-1}(\mathbf{S}_i + \mathbf{S}_j) + \mathbf{I}|}} \exp\left( -\frac{1}{2}(\mu_i - \mu_j)^\top (\mathbf{S}_i + \mathbf{S}_j + \mathbf{K}_{ij})^{-1} (\mu_i - \mu_j) \right) \quad (9)$$

*Proof.* Since $\mathbf{z}_i \sim \mathcal{N}(\mu_i, \mathbf{S}_i)$ and $\mathbf{z}_j \sim \mathcal{N}(\mu_j, \mathbf{S}_j)$ are independent random variables, the difference, $\mathbf{z}_i - \mathbf{z}_j$ is also normally distributed so we can write $\mathbf{z}_i - \mathbf{z}_j \sim \mathcal{N}(\mu_i - \mu_j, \mathbf{S}_i + \mathbf{S}_j)$. In other words, $\mathbf{z}_{ij} \sim \mathcal{N}(\mu_{ij}, \mathbf{S}_{ij})$ where $\mu_{ij} := \mu_i - \mu_j$, $\mathbf{z}_{ij} := \mathbf{z}_i - \mathbf{z}_j$, and $\mathbf{S}_{ij} := \mathbf{S}_i + \mathbf{S}_j$, then we can

write the expected term as follows:

$$\mathbb{E}_{\substack{\mathbf{z}_i \sim \mathcal{N}(\mu_i, \mathbf{S}_i) \\ \mathbf{z}_j \sim \mathcal{N}(\mu_j, \mathbf{S}_j)}} \left[ \exp\left( -\frac{1}{2}(\mathbf{z}_i - \mathbf{z}_j)^\top \mathbf{K}_{ij}^{-1}(\mathbf{z}_i - \mathbf{z}_j) \right) \right] \tag{10}$$

$$= \mathbb{E}_{\mathbf{z}_i - \mathbf{z}_j \sim \mathcal{N}(\mu_{ij}, \mathbf{S}_{ij})} \left[ \exp\left( -\frac{1}{2}(\mathbf{z}_i - \mathbf{z}_j)^\top \mathbf{K}_{ij}^{-1}(\mathbf{z}_i - \mathbf{z}_j) \right) \right] \tag{11}$$

$$= \mathbb{E}_{\mathbf{z}_{ij} \sim \mathcal{N}(\mu_{ij}, \mathbf{S}_{ij})} \left[ \exp\left( -\frac{1}{2}\mathbf{z}_{ij}^\top \mathbf{K}_{ij}^{-1}\mathbf{z}_{ij} \right) \right] \tag{12}$$

$$= \int \frac{1}{\sqrt{(2\pi)^D |\mathbf{S}_{ij}|}} \exp\left( -\frac{1}{2}(\mathbf{z}_{ij} - \mu_{ij})^\top \mathbf{S}_{ij}^{-1}(\mathbf{z}_{ij} - \mu_{ij}) \right) \exp\left( -\frac{1}{2}\mathbf{z}_{ij}^\top \mathbf{K}_{ij}^{-1}\mathbf{z}_{ij} \right) \mathrm{d}\mathbf{z}_{ij} \tag{13}$$

$$= \frac{1}{\sqrt{(2\pi)^D |\mathbf{S}_{ij}|}} \int \exp\left( -\frac{1}{2}\left( (\mathbf{z}_{ij} - \mu_{ij})^\top \mathbf{S}_{ij}^{-1}(\mathbf{z}_{ij} - \mu_{ij}) \right) + \mathbf{z}_{ij}^\top \mathbf{K}_{ij}^{-1}\mathbf{z}_{ij} \right) \mathrm{d}\mathbf{z}_{ij} \tag{14}$$

By expanding and regrouping the terms in the integral, we can write that:

$$\int \exp\left( -\frac{1}{2}\left( (\mathbf{z}_{ij} - \mu_{ij})^\top \mathbf{S}_{ij}^{-1}(\mathbf{z}_{ij} - \mu_{ij}) + \mathbf{z}_{ij}^\top \mathbf{K}_{ij}^{-1}\mathbf{z}_{ij} \right) \right) \mathrm{d}\mathbf{z}_{ij} \tag{15}$$

$$= \int \exp\left( -\frac{1}{2}\left( \mathbf{z}_{ij}^\top \mathbf{K}_{ij}^{-1}\mathbf{z}_{ij} + \mathbf{z}_{ij}^\top \mathbf{S}_{ij}^{-1}\mathbf{z}_{ij} - 2\mathbf{z}_{ij}^\top \mathbf{S}_{ij}^{-1}\mu_{ij} + \mu_{ij}^\top \mathbf{S}_{ij}^{-1}\mu_{ij} \right) \right) \mathrm{d}\mathbf{z}_{ij} \tag{16}$$

$$= \int \exp\left( -\frac{1}{2}\left( \mathbf{z}_{ij}^\top (\mathbf{K}_{ij}^{-1} + \mathbf{S}_{ij}^{-1})\mathbf{z}_{ij} - 2\mathbf{z}_{ij}^\top \mathbf{S}_{ij}^{-1}\mu_{ij} + \mu_{ij}^\top \mathbf{S}_{ij}^{-1}\mu_{ij} \right) \right) \mathrm{d}\mathbf{z}_{ij} \tag{17}$$

$$= \int \exp\left( -\frac{1}{2}\left( \mathbf{z}_{ij}^\top \mathbf{A}_{ij}\mathbf{z}_{ij} - 2\mathbf{z}_{ij}^\top \mathbf{S}_{ij}^{-1}\mu_{ij} + \mu_{ij}^\top \mathbf{S}_{ij}^{-1}\mu_{ij} \right) \right) \mathrm{d}\mathbf{z}_{ij} \tag{18}$$

$$= \int \exp\Bigg( -\frac{1}{2}\Big( (\mathbf{z}_{ij} - \mathbf{A}_{ij}^{-1}\mathbf{S}_{ij}^{-1}\mu_{ij})^\top \mathbf{A}_{ij}(\mathbf{z}_{ij} - \mathbf{A}_{ij}^{-1}\mathbf{S}_{ij}^{-1}\mu_{ij}) -$$
$$\mu_{ij}^\top \mathbf{S}_{ij}^{-1}\mathbf{A}_{ij}^{-1}\mathbf{S}_{ij}^{-1}\mu_{ij} + \mu_{ij}^\top \mathbf{S}_{ij}^{-1}\mu_{ij} \Big) \Bigg) \mathrm{d}\mathbf{z}_{ij} \tag{19}$$

$$= \sqrt{(2\pi)^D |\mathbf{A}_{ij}^{-1}|} \exp\left( -\frac{1}{2}\left( -\mu_{ij}^\top \mathbf{S}_{ij}^{-1}\mathbf{A}_{ij}^{-1}\mathbf{S}_{ij}^{-1}\mu_{ij} + \mu_{ij}^\top \mathbf{S}_{ij}^{-1}\mu_{ij} \right) \right) \mathrm{d}\mathbf{z}_{ij} \tag{20}$$

where $\mathbf{A}_{ij} := \mathbf{K}_{ij}^{-1} + \mathbf{S}_{ij}^{-1}$. In Eq. 19, we add and substract the term $\mu_{ij}^\top \mathbf{S}_{ij}^{-1}\mathbf{A}_{ij}^{-1}\mathbf{S}_{ij}^{-1}\mu_{ij}$ so that the first component depends only on $\mathbf{z}_{ij}$. It also corresponds to the numerator of a normal distribution with mean $\mathbf{A}_{ij}^{-1}\mathbf{S}_{ij}^{-1}\mu_{ij}$ and covariance $\mathbf{A}_{ij}^{-1}$. Hence, the last equality follows from the standard Gaussian integral:

$$\int \exp\left( -\frac{1}{2}(\mathbf{z}_{ij} - \mathbf{A}_{ij}^{-1}\mathbf{S}_{ij}^{-1}\mu_{ij})^\top \mathbf{A}_{ij}(\mathbf{z}_{ij} - \mathbf{A}_{ij}^{-1}\mathbf{S}_{ij}^{-1}\mu_{ij}) \right) \mathrm{d}\mathbf{z}_{ij} = \sqrt{(2\pi)^D |\mathbf{A}_{ij}^{-1}|}$$

Therefore, the expectation term in Eq. 12 can then be rewritten as follows:

$$\mathbb{E}_{\substack{\mathbf{z}_i \sim \mathcal{N}(\mu_i, \mathbf{S}_i) \\ \mathbf{z}_j \sim \mathcal{N}(\mu_j, \mathbf{S}_j)}} \left[ \exp\left( -\frac{1}{2}(\mathbf{z}_i - \mathbf{z}_j)^\top \mathbf{K}_{ij}^{-1}(\mathbf{z}_i - \mathbf{z}_j) \right) \right] \tag{21}$$

$$= \frac{1}{\sqrt{(2\pi)^D |\mathbf{S}_{ij}|}} \int \exp\left( -\frac{1}{2}\left( 2\mathbf{z}_{ij}^\top \mathbf{K}_{ij}^{-1}\mathbf{z}_{ij} + (\mathbf{z}_{ij} - \mu_{ij})^\top \mathbf{S}_{ij}^{-1}(\mathbf{z}_{ij} - \mu_{ij}) \right) \right) \mathrm{d}\mathbf{z}_{ij} \tag{22}$$

$$= \frac{1}{\sqrt{(2\pi)^D |\mathbf{S}_{ij}|}} \sqrt{(2\pi)^D |\mathbf{A}_{ij}^{-1}|} \exp\left( -\frac{1}{2}\left( -\mu_{ij}^\top \mathbf{S}_{ij}^{-1}\mathbf{A}_{ij}^{-1}\mathbf{S}_{ij}^{-1}\mu_{ij} + \mu_{ij}^\top \mathbf{S}_{ij}^{-1}\mu_{ij} \right) \right) \tag{23}$$

$$= \frac{1}{\sqrt{|\mathbf{A}_{ij}||\mathbf{S}_{ij}|}} \exp\left( -\frac{1}{2}\left( -\mu_{ij}^\top \mathbf{S}_{ij}^{-1}\mathbf{A}_{ij}^{-1}\mathbf{S}_{ij}^{-1}\mu_{ij} + \mu_{ij}^\top \mathbf{S}_{ij}^{-1}\mu_{ij} \right) \right) \tag{24}$$

where $|\mathbf{A}_{ij}^{-1}| = |\mathbf{A}_{ij}|^{-1}$. By substituting $\mathbf{A}_{ij}$ with $(\mathbf{K}_{ij}^{-1} + \mathbf{S}_{ij}^{-1})$ and applying the Woodbury Matrix Identity, we can obtain

$$
\mathbb{E}_{\substack{\mathbf{z}_i \sim \mathcal{N}(\mu_i, \mathbf{S}_i) \\ \mathbf{z}_j \sim \mathcal{N}(\mu_j, \mathbf{S}_j)}} \left[ \exp\left( -\frac{1}{2}(\mathbf{z}_i - \mathbf{z}_j)^\top \mathbf{K}_{ij}^{-1}(\mathbf{z}_i - \mathbf{z}_j) \right) \right] \tag{25}
$$

$$
= \frac{1}{\sqrt{|\mathbf{A}_{ij}||\mathbf{S}_{ij}|}} \exp\left( -\frac{1}{2}\left( -\mu_{ij}^\top \mathbf{S}_{ij}^{-1} \mathbf{A}_{ij}^{-1} \mathbf{S}_{ij}^{-1} \mu_{ij} + \mu_{ij}^\top \mathbf{S}_{ij}^{-1} \mu_{ij} \right) \right) \tag{26}
$$

$$
= \frac{1}{\sqrt{|\mathbf{K}_{ij}^{-1} + \mathbf{S}_{ij}^{-1}||\mathbf{S}_{ij}|}} \exp\left( -\frac{1}{2}\left( -\mu_{ij}^\top \mathbf{S}_{ij}^{-1}(\mathbf{K}_{ij}^{-1} + \mathbf{S}_{ij}^{-1})^{-1} \mathbf{S}_{ij}^{-1} \mu_{ij} + \mu_{ij}^\top \mathbf{S}_{ij}^{-1} \mu_{ij} \right) \right) \tag{27}
$$

$$
= \frac{1}{\sqrt{|\mathbf{K}_{ij}^{-1}\mathbf{S}_{ij} + \mathbf{I}|}} \exp\left( -\frac{1}{2}\mu_{ij}^\top \left( \mathbf{S}_{ij}^{-1} - \mathbf{S}_{ij}^{-1}(\mathbf{K}_{ij}^{-1} + \mathbf{S}_{ij}^{-1})^{-1} \mathbf{S}_{ij}^{-1} \right) \mu_{ij} \right) \tag{28}
$$

$$
= \frac{1}{\sqrt{|\mathbf{K}_{ij}^{-1}\mathbf{S}_{ij} + \mathbf{I}|}} \exp\left( -\frac{1}{2}\mu_{ij}^\top \left( \mathbf{S}_{ij} + \mathbf{K}_{ij} \right)^{-1} \mu_{ij} \right) \tag{29}
$$

Finally, by substituting the terms, $\mu_{ij}$ and $\mathbf{S}_{ij}$, we can conclude that

$$
\mathbb{E}_{\substack{\mathbf{z}_i \sim \mathcal{N}(\mu_i, \mathbf{S}_i) \\ \mathbf{z}_j \sim \mathcal{N}(\mu_j, \mathbf{S}_j)}} \left[ \exp\left( -\frac{1}{2}(\mathbf{z}_i - \mathbf{z}_j)^\top \mathbf{K}_{ij}^{-1}(\mathbf{z}_i - \mathbf{z}_j) \right) \right] \tag{30}
$$

$$
= \frac{1}{\sqrt{|\mathbf{K}_{ij}^{-1}(\mathbf{S}_i + \mathbf{S}_j) + \mathbf{I}|}} \exp\left( -\frac{1}{2}(\mu_i - \mu_j)^\top \left( \mathbf{S}_i + \mathbf{S}_j + \mathbf{K}_{ij} \right)^{-1} (\mu_i - \mu_j) \right) \tag{31}
$$

$\square$

**Lemma A.2.** *Let $\epsilon \in (0, 1/2)$, and let $\phi : \mathcal{S} \to \mathbb{R}^D \times \mathbb{R}_+^D$ be an $\epsilon$-distinguishable embedding function for a pair $(\mathbf{s}_i, \mathbf{s}_j) \in \mathcal{S}^2$ and label $y_{ij} \in \{0, 1\}$ where $\phi := (\phi_\mu, \phi_\sigma)$ with $\phi_\mu : \mathcal{S} \to \mathbb{R}^D$ and $\phi_\sigma : \mathcal{S} \to \mathbb{R}_+^D$. Then the following bounds hold:*

$$
\min_d \{(\phi_\sigma(\mathbf{s}_i) + \phi_\sigma(\mathbf{s}_j))_d\} \log\left( \frac{1}{\epsilon^4} \right) \le \|\phi_\mu(\mathbf{s}_i) - \phi_\mu(\mathbf{s}_j)\|_2^2 \qquad \text{if } y_{ij} = 0, \tag{32}
$$

$$
\max_d \{(\phi_\sigma(\mathbf{s}_i) + \phi_\sigma(\mathbf{s}_j))_d\} \log\left( \frac{1}{(1-\epsilon)^4} \right) \ge \|\phi_\mu(\mathbf{s}_i) - \phi_\mu(\mathbf{s}_j)\|_2^2 \qquad \text{if } y_{ij} = 1. \tag{33}
$$

*Proof.* Let us define $\sigma^2 := \phi_\sigma(\mathbf{s}_i) + \phi_\sigma(\mathbf{s}_j)$ and $\mu = \phi_\mu(\mathbf{s}_i) - \phi_\mu(\mathbf{s}_j)$. We first establish the lower bound for a negative pair ($y_{ij} = 0$). By $\epsilon$-distinguishability, we assume that $q(Y_{ij} = 0 \mid \cdots) \ge 1 - \epsilon$ then $\exp\left( -\frac{1}{4} \mu^\top \mathbf{S}_{ij}^{-1} \mu \right) \le \epsilon$, where $\mathbf{S}_{ij} := \mathrm{diag}(\sigma^2)$. Hence, $\mu^\top \mathbf{S}_{ij}^{-1} \mu \ge -4 \log(\epsilon)$, and we can obtain

$$
\min_d \{\sigma_d^2\} \log\left( \epsilon^{-4} \right) \le \min_d \{\sigma_d^2\} \mu^\top \mathbf{S}_{ij}^{-1} \mu \tag{34}
$$

$$
= \min_d \{\sigma_d^2\} \sum_{d=1}^D \frac{\mu_d^2}{\sigma_d^2} \tag{35}
$$

$$
\le \min_d \{\sigma_d^2\} \left( \frac{1}{\min_d \{\sigma_d^2\}} \sum_{d=1}^D \mu_d^2 \right) \tag{36}
$$

$$
= \|\phi_\mu(\mathbf{s}_i) - \phi_\mu(\mathbf{s}_j)\|_2^2. \tag{37}
$$

Since $\epsilon \in (0, 1/2)$, the left-hand side is strictly positive, giving a nontrivial lower bound.

For a positive pair ($y_{ij} = 1$), similarly, we have $\exp\left(-\frac{1}{4}\mu^\top \mathbf{S}_{ij}^{-1}\mu\right) \geq 1-\epsilon$, which implies $\mu^\top \mathbf{S}_{ij}^{-1}\mu \leq 4\log\left(\frac{1}{1-\epsilon}\right)$. Then, we can write

$$\max_d\{\sigma_d^2\} \log\left(\frac{1}{(1-\epsilon)^4}\right) \geq \max_d\{\sigma_d^2\}\,\mu^\top \mathbf{S}_{ij}^{-1}\mu \tag{38}$$

$$= \max_d\{\sigma_d^2\} \sum_{d=1}^{D} \frac{\mu_d^2}{\sigma_d^2} \tag{39}$$

$$\geq \max_d\{\sigma_d^2\} \left(\frac{1}{\max_d\{\sigma_d^2\}} \sum_{d=1}^{D} \mu_d^2\right) \tag{40}$$

$$= \sum_{d=1}^{D} \mu_d^2 \tag{41}$$

$$= \|\phi_\mu(\mathbf{s}_i) - \phi_\mu(\mathbf{s}_j)\|_2^2. \tag{42}$$

This establishes the upper bound for the embedding distances when $y_{ij} = 1$. $\qquad\square$

**Lemma A.3.** *Let $\tau > 0$ and $\mathbf{z}_0 \in \mathbb{R}^D$. Then, there exist at most $\mathcal{P}_\tau^D$ distinct points $\{\mathbf{z}_i\}_{i\geq 1} \subset \mathbb{R}^D$ such that*

$$\|\mathbf{z}_i - \mathbf{z}_0\| \leq \tau \qquad and \qquad \|\mathbf{z}_i - \mathbf{z}_j\| \geq \tau \quad for\ all\ 1 \leq i < j \leq \mathcal{P}_\tau^D, \tag{43}$$

*where $\mathcal{P}_\tau^D$ is the* packing number *of a unit Euclidean ball in $\mathbb{R}^D$.*

*Proof.* By definition, the packing number, $\mathcal{P}_\tau^D$, is the maximal number of points that can fit in $B(\mathbf{z}_0, \tau)$ such that any two points are at least $\tau$ apart. Therefore, any set of points in $B(\mathbf{z}_0, \tau)$ satisfying $\|\mathbf{z}_i - \mathbf{z}_j\| \geq \tau$ can contain at most $\mathcal{P}_\tau^D$ points. $\qquad\square$

**Corollary A.3.1.** *Let $\phi : \mathcal{S} \to \mathbb{R}^D \times \mathbb{R}_+^D$ be an embedding function for the set $\mathcal{S}$ where $\phi := (\phi_\mu, \phi_\sigma)$ with $\phi_\mu : \mathcal{S} \to \mathbb{R}^D$ and $\phi_\sigma : \mathcal{S} \to \mathbb{R}_+^D$. If $\phi_\sigma(\mathbf{s}_i)_d = \phi_\sigma(\mathbf{s}_j)_d$ for all $\mathbf{s}_i, \mathbf{s}_j \in \mathcal{S}$, and $\forall d \in [D]$, and if there exists $\mathcal{P}_\tau^D + 2$ sequences, $\mathbf{s}_0, \mathbf{s}_1, \ldots, \mathbf{s}_{\mathcal{P}_\tau^D + 1} \in \mathcal{S}$ such that each $(\mathbf{s}_0, \mathbf{s}_i)$ is a positive pair (i.e. $y_{(0,i)} = 1$) for all $i \in \{1, \ldots \mathcal{P}_\tau^D + 1\}$ and $(\mathbf{s}_i, \mathbf{s}_j)$ is a negative pair (i.e. $y_{(i,j)} = 0$) for $1 \leq i < j \leq \mathcal{P}^D + 1$, then it cannot be $\epsilon$-distinguishable function for $\epsilon \in (0, 1/2)$.*

*Proof.* Assume for contradiction that such an $\epsilon$-distinguishable function $\phi : \mathcal{S} \to \mathbb{R}^D \times \mathbb{R}_+^D$ exists for some $\epsilon \in (0, 1/2)$ with constant variance terms $\sigma^2 := \phi_\sigma(\mathbf{s}_i)_d = \phi_\sigma(\mathbf{s}_j)_d$ for all $(\mathbf{s}_i, \mathbf{s}_j) \in \mathcal{S}^2$ and $d \in [D]$ so Lemma 3.3 implies that

$$\sigma^2 \log\left(\frac{1}{\epsilon^4}\right) \leq \|\phi_\mu(\mathbf{s}_i) - \phi_\mu(\mathbf{s}_j)\|_2^2 \qquad \text{if } y_{ij} = 0 \tag{44}$$

$$\sigma^2 \log\left(\frac{1}{(1-\epsilon)^4}\right) \geq \|\phi_\mu(\mathbf{s}_i) - \phi_\mu(\mathbf{s}_j)\|_2^2 \qquad \text{if } y_{ij} = 1. \tag{45}$$

for all $(\mathbf{s}_i, \mathbf{s}_j) \in \mathcal{S}$ pairs. Note that we have

$$\log\left(\frac{1}{\epsilon^4}\right) - \log\left(\frac{1}{(1-\epsilon)^4}\right) = \log\left(\frac{(1-\epsilon)^4}{\epsilon^4}\right) = 4\log\left(\epsilon^{-1} - 1\right) > 0 \tag{46}$$

so let's define $\tau_\epsilon := \left(\log(\epsilon^{-4}) + \log((1-\epsilon)^{-4})\right)/2$, then we can write

$$\|\phi_\mu(\mathbf{s}_i) - \phi_\mu(\mathbf{s}_j)\|_2 > \sigma\sqrt{\tau_\epsilon}, \tag{47}$$

for all negative pairs $(\mathbf{s}_i, \mathbf{s}_j)$ $1 \leq i < j \leq \mathcal{P}^D + 1$ and we have

$$\|\phi_\mu(\mathbf{s}_0) - \phi_\mu(\mathbf{s}_i)\|_2 < \sigma\sqrt{\tau_\epsilon} \tag{48}$$

for all positive pairs $(\mathbf{s}_0, \mathbf{s}_i)$ where $1 \leq i \leq \mathcal{P}_\tau^D + 1$. In other words, each $\phi_\mu(\mathbf{s}_i)$ ($i \geq 1$) lies within a ball of radius $\sigma\sqrt{\tau_\epsilon}$ centered at $\phi_\mu(\mathbf{s}_0)$. However, the condition in Eq. 47 requires that all negative pairs have to be at least $\sigma\sqrt{\tau_\epsilon}$ apart from each other at the same time but the maximum number of points that are $\sigma\sqrt{\tau_\epsilon}$ apart in $B(\phi_\mu(\mathbf{s}_0), \sigma\sqrt{\tau_\epsilon})$ is at most $\mathcal{P}_\tau^D$. Therefore, we obtain a contradiction. $\qquad\square$

**Theorem A.4.** *An embedding function $\phi : \mathcal{S} \to \mathbb{R}^D \times \mathbb{R}_+^D$ with bounded means (i.e. $\|\phi_\mu(\mathbf{s})\| < \infty$) is $\epsilon$-distinguishable for some $\epsilon \in (0, 1/2)$ if and only if there exists a set of sequences $\{\mathbf{s}_0, \mathbf{s}_1, \ldots, \mathbf{s}_N\} \subseteq \mathcal{S}$ where each $(\mathbf{s}_0, \mathbf{s}_i)$ is a positive pair and $(\mathbf{s}_i, \mathbf{s}_j)$ is negative pair satisfying $\phi_\sigma(\mathbf{s}_i)_d < \infty$ for $1 \le i \le N$ and $\phi_\sigma(\mathbf{s}_0)_d \to \infty$ for all $d \in [D]$ with $N > \mathcal{P}_\tau^D$.*

*Proof.* Suppose there exists such a set of sequences $\mathbf{s}_0, \mathbf{s}_1, \ldots, \mathbf{s}_N \in \mathcal{S}$ for $N > \mathcal{P}_\tau^D$. Since each $(\mathbf{s}_0, \mathbf{s}_i)$ is a positive pair for $1 \le i \le N$, we have

$$\log\left((1-\epsilon)^{-4}\right) \ge \left(\phi_\mu(\mathbf{s}_0) - \phi_\mu(\mathbf{s}_i)\right)^\top \left(\mathbf{S}_0 + \mathbf{S}_i\right)^{-1} \left(\phi_\mu(\mathbf{s}_0) - \phi_\mu(\mathbf{s}_i)\right) \tag{49}$$

where $\mathbf{S}_0 := \mathrm{diag}(\phi_\sigma(\mathbf{s}_0))$ and $\mathbf{S}_i := \mathrm{diag}(\phi_\sigma(\mathbf{s}_i))$. Similarly, for a negative pair $(\mathbf{s}_i, \mathbf{s}_j)$, we can write

$$\left(\phi_\mu(\mathbf{s}_i) - \phi_\mu(\mathbf{s}_j)\right)^\top \left(\mathbf{S}_i + \mathbf{S}_j\right)^{-1} \left(\phi_\mu(\mathbf{s}_i) - \phi_\mu(\mathbf{s}_j)\right) \ge \log\left(\epsilon^{-4}\right) \tag{50}$$

for $1 \le i < j \le N$. Note that $\log\left(\epsilon^{-4}\right) > \log\left((1-\epsilon)^{-4}\right)$ for $\epsilon \in (0, 1/2)$ so it implies that

$$\frac{\left(\phi_\mu(\mathbf{s}_0) - \phi_\mu(\mathbf{s}_i)\right)^\top \left(\mathbf{S}_0 + \mathbf{S}_i\right)^{-1} \left(\phi_\mu(\mathbf{s}_0) - \phi_\mu(\mathbf{s}_i)\right)}{\left(\phi_\mu(\mathbf{s}_i) - \phi_\mu(\mathbf{s}_j)\right)^\top \left(\mathbf{S}_i + \mathbf{S}_j\right)^{-1} \left(\phi_\mu(\mathbf{s}_i) - \phi_\mu(\mathbf{s}_j)\right)} \le \frac{\log\left((1-\epsilon)^{-4}\right)}{\log\left(\epsilon^{-4}\right)} \to 0 \quad \text{as} \quad \epsilon \to 0 \tag{51}$$

Since the embeddings are bounded, i.e. $\|\phi_\mu(\mathbf{s})\|_2 < \infty \ \forall \mathbf{s} \in \mathcal{S}$, then either $(\phi_\sigma(\mathbf{s}_i)_d + \phi_\sigma(\mathbf{s}_j)_d) \to 0^+$ holds for some $d \in [D]$ or $(\phi_\sigma(\mathbf{s}_0)_d + \phi_\sigma(\mathbf{s}_i)_d) \to \infty$ for every $d \in [D]$. The first case cannot happen by Lemma A.3, and Corollary A.3.1 so $\phi_\sigma(\mathbf{s}_0)_d \to \infty^+$ for every $d \in [D]$.

For the other direction of the statement, assume that $\phi_\sigma(\mathbf{s}_i)_d < \infty$ and $1 \le i \le N$ and $\phi_\sigma(\mathbf{s}_0)_d \to \infty$ for every $d \in [D]$. Then, for a given $\epsilon \in (0, 1/2)$, we can find $M_\epsilon$ such that $\min_d\{(\phi_\sigma(\mathbf{s}_0)_d + \phi_\sigma(\mathbf{s}_i)_d)\} \ge M_\epsilon \ge \frac{1}{4\epsilon}\|\phi_\sigma(\mathbf{s}_0 - \phi_\sigma(\mathbf{s}_i)\|_2^2$, and it implies that $\epsilon > \frac{1}{4}\frac{\|\phi_\mu(\mathbf{s}_0) - \phi_\mu(\mathbf{s}_i)\|_2^2}{\min_d\{(\phi_\sigma(\mathbf{s}_0)_d + \phi_\sigma(\mathbf{s}_i)_d)\}} \ge \frac{1}{4}\sum_{d=1}^D \frac{(\phi_\mu(\mathbf{s}_0) - \phi_\mu(\mathbf{s}_i))_d^2}{(\phi_\sigma(\mathbf{s}_0) + \phi_\sigma(\mathbf{s}_i))_d}$ so

$$1 - \epsilon \le \left(1 - \frac{1}{4}\sum_{d=1}^D \frac{\left(\phi_\mu(\mathbf{s}_0)_d - \phi_\mu(\mathbf{s}_i)_d\right)^2}{\left(\phi_\sigma(\mathbf{s}_0)_d + \phi_\sigma(\mathbf{s}_i)_d\right)}\right) \tag{52}$$

$$\le \exp\left(-\frac{1}{4}\sum_{d=1}^D \frac{\left(\phi_\mu(\mathbf{s}_0)_d - \phi_\mu(\mathbf{s}_i)_d\right)^2}{\left(\phi_\sigma(\mathbf{s}_0)_d + \phi_\sigma(\mathbf{s}_i)_d\right)}\right) \tag{53}$$

$$= \exp\left(-\frac{1}{4}\left(\phi_\mu(\mathbf{s}_0) - \phi_\mu(\mathbf{s}_i)\right)^\top \left(\mathbf{S}_0 + \mathbf{S}_i\right)^{-1}\left(\phi_\mu(\mathbf{s}_0) - \phi_\mu(\mathbf{s}_i)\right)\right) \tag{54}$$

$$= \beta\mathbb{E}_{\substack{\mathbf{z}_i \sim \mathcal{N}(\mu_i, \mathbf{S}_i) \\ \mathbf{z}_j \sim \mathcal{N}(\mu_j, \mathbf{S}_j)}}\left[p(y_{(i,j)} = 1 \mid \cdots)\right] \tag{55}$$

$$= q(Y_{ij} = 1 \mid \cdots) \tag{56}$$

and the second line follows from the inequality $1 - x \le \exp(-x)$ for all $x \in \mathbb{R}$.

For negative pairs, similarly, let $M_\epsilon := \max_d\{\phi_\sigma(\mathbf{s}_i)_d + \phi_\sigma(\mathbf{s}_j)_d\} \le \frac{D}{4\log(1/\epsilon)}\|\phi_\mu(\mathbf{s}_i) - \phi_\mu(\mathbf{s}_j)\|_2^2$ for all $1 \le i < j \le N$.

$$1 - \epsilon = 1 - \exp(-\log(1/\epsilon)) \tag{57}$$

$$= 1 - \exp\left(-\frac{1}{4}\sum_{d=1}^D \frac{4\log(1/\epsilon)}{D}\right) \tag{58}$$

$$\le 1 - \exp\left(-\frac{1}{4}\frac{1}{\max_d\{\phi_\sigma(\mathbf{s}_i)_d + \phi_\sigma(\mathbf{s}_j)_d\}}\|\phi_\mu(\mathbf{s}_i)_d - \phi_\mu(\mathbf{s}_j)_d\|_2^2\right) \tag{59}$$

$$\le 1 - \exp\left(-\frac{1}{4}\sum_{d=1}^D \frac{\left(\phi_\mu(\mathbf{s}_i)_d - \phi_\mu(\mathbf{s}_j)_d\right)^2}{\left(\phi_\sigma(\mathbf{s}_i)_d + \phi_\sigma(\mathbf{s}_j)_d\right)}\right) \tag{60}$$

$$= 1 - \exp\left(-\frac{1}{4}\left(\phi_\mu(\mathbf{s}_i) - \phi_\mu(\mathbf{s}_j)\right)^\top \left(\mathbf{S}_i + \mathbf{S}_j\right)^{-1}\left(\phi_\mu(\mathbf{s}_i) - \phi_\mu(\mathbf{s}_j)\right)\right) \tag{61}$$

$$= 1 - \beta\mathbb{E}_{\substack{\mathbf{z}_i \sim \mathcal{N}(\mu_i, \mathbf{S}_i) \\ \mathbf{z}_j \sim \mathcal{N}(\mu_j, \mathbf{S}_j)}}\left[p(y_{(i,j)} = 1 \mid \cdots)\right] \tag{62}$$

$$= q(Y_{ij} = 0 \mid \cdots) \tag{63}$$

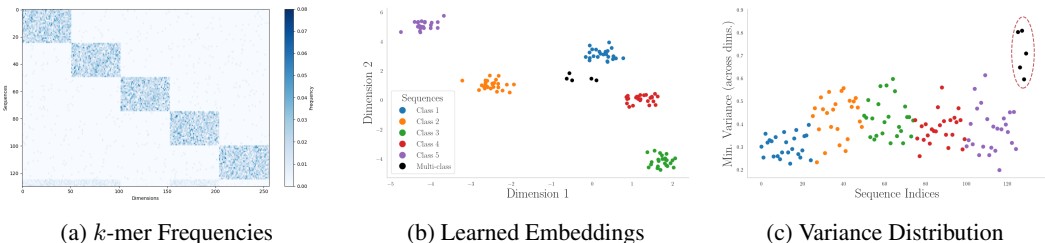

|  (a) $k$-mer Frequencies | (b) Learned Embeddings | (c) Variance Distribution |

Figure 6: Visualization of the input $k$-mer features with learned embeddings and the variances.

Therefore, the function $\phi : \mathcal{S} \to \mathbb{R}^D \times \mathbb{R}^D_+$ is an $\epsilon$-distinguishable embedding function.

□

## B    EXPERIMENTS

**Datasets.** For our experiments, we adopt the benchmark datasets introduced in prior work on the metagenomic binning task (Zhou et al., 2024). The datasets are constructed from reference genomes in GenBank and consist of viral, fungal, and bacterial sequences. The training data contains more than 2 million sequence pairs of length 1000bp. For testing, we have six datasets (*Reference 5/6*, *Plant 5/6*, and *Marine 5/6*) with species represented by highly variable numbers of sequences (10–4, 599), ranging from 2-20 kbp in length. While *Reference* datasets consist of DNA fragments from 250-330 fungal and viral genomes, and *Marine* and *Plant*-associated environments contain 70k-125k sequences from roughly 180-520 species.

**Baselines.** KMER(COSINE) and KMER($\ell_1$) are the classical representations based on 4-mer frequencies, where sequence similarity is computed using either cosine similarity or an exponential kernel over the $\ell_1$ distance. HYENADNA (Nguyen et al., 2023) is a genome foundation model, operating at single-nucleotide resolution with context lengths up to $10^6$ to efficiently capture long-range dependencies beyond the reach of standard transformers. DNABERT-2 (Zhou et al., 2023) is also a foundation model that replaces fixed $k$-mer tokenization with Byte Pair Encoding (BPE) to improve modeling efficiency. DNABERT-S (Zhou et al., 2024) leverages DNABERT-2 as a pretrained backbone, fine-tuned with contrastive objectives tailored to metagenomic binning. RE-VISTKMERS (Çelikkanat et al., 2024) is a lightweight model that learns sequence embeddings via a two-layer neural network applied to 4-mer profiles. It provides a strong deterministic baseline and can be viewed as the non-probabilistic counterpart of our approach. REVISTKMERS is thus also the main baseline in the experimental setup as the proposed model shares the lightweight characteristics of REVISTKMERS.

### B.1    TOY EXAMPLE WITH $k$-MER DATASET

To evaluate the behavior of our model in a controlled setting, we designed a synthetic toy dataset of $k$-mer sequences. The goal of this study is twofold: (i) to verify that the model can learn meaningful low-dimensional embeddings that reflect the underlying class structure, and (ii) to assess how the model represents sequences belonging to multiple classes with their mean and variance terms.

The dataset consists of sequences generated from multinomial distributions with a clear class structure, enabling us to systematically analyze the effect of sequence overlap and class separability on the learned embeddings.

**Data Generation.** We generated sequences of length 100 composed of 4-mers, resulting in 256 possible $k$-mer types. The sequences were then divided into 5 classes, each containing 25 sequences. For each class, a multinomial distribution was defined such that the probability mass was concentrated on a distinct subset of k-mer dimensions, with a small uniform smoothing $10^{-2}$ to avoid zero probabilities. It ensured that sequences within the same class had similar $k$-mer compositions, while sequences from different classes were distinguishable (Figure 6a). To further test the model's ability to handle ambiguity, we introduced 5 "multi-class" sequences sampled by combining $k$-mer counts

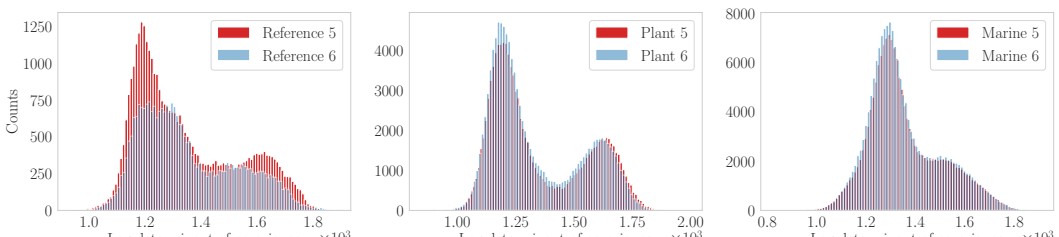

Figure 7: Distribution of the log-determinant of covariance matrices across datasets.

from multiple classes. This design simulates sequences that do not belong exclusively to a single class and allows us to probe how the model handles overlapping class structures.

**Pair construction.** We constructed positive sequence pairs by sampling two sequences from the same class, effectively assuming full access to positive examples (i.e., $y_{ij} = 1$). Negative pairs were generated using the same random sampling strategy described in Section 4, which naturally introduces the possibility of false negatives. For the multi-class sequences, we additionally constructed pairs with sequences from their contributing classes, enabling the model to learn representations that account for both pure and mixed class memberships.

**Results.** We learn sequence embeddings directly in a 2-dimensional latent space (Figure 6b), avoiding any dimensionality reduction steps that could distort the geometric structure of the embedding space. Visualization of the learned embeddings reveals that sequences from distinct classes form well-separated groups, while sequences belonging to multiple classes occupy intermediate regions. Importantly, our model also predicts a diagonal covariance matrix for each sequence, capturing the uncertainty associated with sequences that span multiple clusters.

By Lemma 3.3 and Theorem 3.4, the minimum variance across dimensions provides a lower bound on pairwise distances. Therefore, we expect multi-class sequences to exhibit larger variance values. Figure 6c illustrates the distribution of $\min_d\{\phi_\sigma(\mathbf{s})_d\}$ for each sequence $\mathbf{s} \in \mathcal{S}$. As predicted by our theoretical analysis, sequences associated with multiple classes indeed show larger minimum variances across dimensions, reflecting their position between clusters in the latent space.

## B.2 ABLATION STUDIES

**Distribution of variances.** We begin our uncertainty analysis by inspecting the distribution of the predicted variance terms. Recall that for a given sequence $\mathbf{s} \in \mathcal{S}$, the model produces dimension-wise variance estimates $\phi_\sigma(\mathbf{s}) \in \mathbb{R}^D_{\geq 0}$ corresponding to the diagonal entries of the covariance matrix. We compute

$$u(\mathbf{s}) = \sum_{d=1}^{D} \log\left(\phi_\sigma(\mathbf{s})_d + 1\right), \tag{64}$$

which is in fact the log-determinant of the covariance matrix, and we have the $+1$ term in order to ensure numerical stability. Therefore, it captures the sequence-level dispersion in the embedding space.

Figure 7 shows the empirical distribution of $u(\mathbf{s})$ over sequences in the testing datasets. The multimodal distributions of $u(\mathbf{s})$ are very clear for *Reference 5/6* and *Plant 5/6* datasets. This indicates that the model partitions the sequence space into distinct regimes of predictive uncertainty, which might point out the two distinct sets of species (Please see Section 4 for the dataset details.). Importantly, the observed distributions are neither degenerate (collapsed near zero) nor uniform. Instead, they encode structured variability that reflects properties of the underlying data distribution. This finding supports the hypothesis that the variance terms carry semantically meaningful information rather than merely acting as nuisance parameters. Hence, these results establish that our model produces non-trivial and interpretable uncertainty estimates.

**Sequence Filtering Across Datasets.** To further probe the role of predictive uncertainty, we conduct a sequence filtering experiment in which we selectively remove sequences with the largest variance

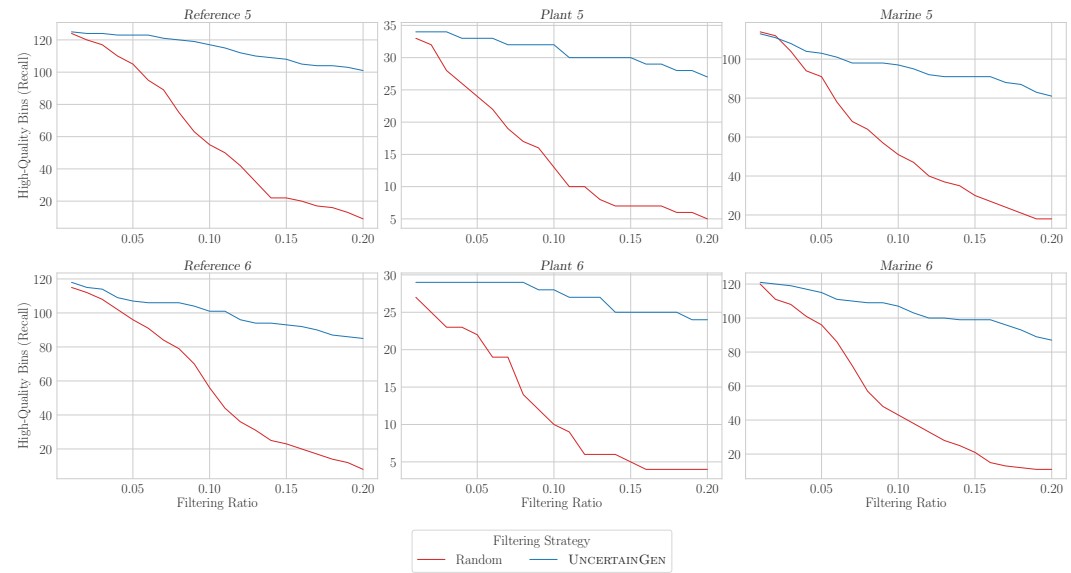

Figure 8: Filtering sequences with varying ratios over all testing datasets.

scores. Specifically, we sort sequences by their aggregated log-determinant values, $u(\mathbf{s})$ (as defined in Eq. 64,) and iteratively filter out the highest-uncertainty items. After filtering, we evaluate cluster quality by reporting the number of clusters that achieve a recall score $\geq 0.9$. For comparison, we include a *random filtering* baseline in which the same number of sequences is removed uniformly at random. To ensure a fair comparison, all removed sequences are assigned to a dedicated "garbage" cluster so that they do not artificially inflate false negatives.

Figure 8 summarizes results for all the benchmarks, and we observe the consistent trends across all datasets. Filtering by predictive uncertainty consistently yields a larger number of clusters with recall $\geq 0.9$ compared to the random baseline. This suggests that uncertainty-guided filtering preserves the integrity of high-quality clusters while selectively removing sequences that would otherwise degrade cluster purity. The sequences assigned the highest variance values might typically originate from low-quality or noisy bins. These sequences might also tend to coincide with cases that contribute to false negatives in the unfiltered setting. By removing them, the model effectively reduces noise in the evaluation and highlights clusters that better reflect true structure in the data. Therefore, the model's variance estimates can serve as a practical mechanism for uncertainty-aware sequence selection and downstream decision making.

**Effect of embedding dimension size.** As discussed in our theoretical analysis (Section 3), incorporating covariance terms provides the model with additional representational capacity compared to squared Euclidean distance: beyond capturing predictive uncertainty, the embedding space approximates a non-Euclidean Riemannian manifold. This enriched geometry has the potential to separate complex sequence structures more effectively, particularly when the embedding dimension is small and representational bottlenecks are most restrictive.

Figure 9) demonstrates this effect on the *Reference* dataset very clearly, where covariance-aware embeddings consistently outperformed the Euclidean baseline. The performance gains were especially pronounced in low-dimensional regimes, aligning well with our theoretical motivation. For the *Plant* and *Marine* datasets, the magnitude of the improvement is marginal and approaches the range of experimental noise. Hence, these results suggest that while covariance terms indeed enrich the representational geometry in a theoretically appealing way and can yield measurable improvements in practice, the empirical benefits are not universal across datasets. Instead, the extent of improvement appears to be dataset-dependent, reflecting differences in the underlying structure and complexity of the sequences.

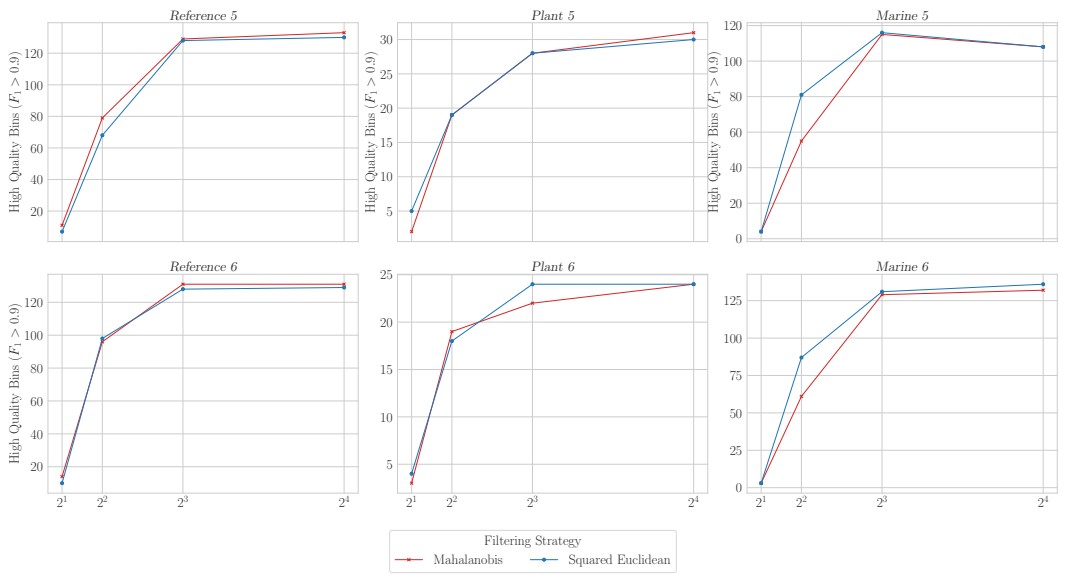

Figure 9: Impact of dimension size for different metrics.

