# OpenReview forum: "UncertainGen: Uncertainty-Aware Representations of DNA Sequences for Metagenomic Binning"
_ICLR.cc/2026/Conference — Submitted to ICLR 2026_

### Official Review · Reviewer_GghP · 2025-10-20

**Soundness:** 3
**Presentation:** 3
**Contribution:** 2
**Rating:** 4
**Confidence:** 3

**Summary:**

Metagenomics binning is an important step for analyzing microbial communities. It can benefit from a probabilistic clustering approach owing to domain-specific caveats such as inter-species DNA sharing. However existing data-driven approaches (e.g. kmer profiles and foundation model embeddings) produce deterministic embeddings.

To address this, the authors propose a probabilistic formulation of contrastive learning, where an encoder produces a mean and variance of a latent representation (similar to that of an inference network in classical VAE literature), and a non-Euclidian similarity objective is optimized over (paired) fragmented sequences. Experiments are conducted over established binning benchmarks, showing that a small, trained-from-scratch MLP can be competitive against a much larger, pretrained foundation model (DNABERT-S).

**Strengths:**

- Metagenomics binning is a well-motivated scientific problem with important applications, and it's good to see efforts designed at improving this task.
- I really appreciate the authors' clarity in presenting their main theoretical results, which very clearly demonstrate the impossibility of deterministic embedding in producing the correct clustering, and how a probabilistic approach could fare better.
- Experimental results are good. A small MLP trained from scratch can match and/or outperform a much larger DNABERT-S finetuned from a capable foundation model (DNABERT-2)

**Weaknesses:**

- I do find the application area to be too siloed in metagenomics applications. While the probabilistic embedding framework is general, the authors should discuss how their approach can be beneficial for the broader scientific ML community, and / or the theoretical contribution is significant / contains deep technical innovations, discussed next.
- Overall, while the theoretical results are neat and clearly presented, they are primarily derived from existing techniques / known results in probability.
- On the empirical side, it is good that a small network can outperform specialized foundation models (DNABERT-S), but it does make the scale of experiments a bit lacking. For example, the authors can finetune DNABERT-S with their objective and evaluate whether this objective can bring meaningful improvements from a strong base model.
- On the other hand, recent works (e.g. [1]) have shown that carefully trained supervised baselines can outperform specialized foundation models in the genomics domain, and it'd be great if the authors can conduct an experiment on their simple MLP, but with the standard contrastive learning objective, and evaluate the model performance on binning benchmarks. This should give us a concrete idea of their objective's benefits.

Overall, I like this work but find it to be a bit too niche for ICLR.

[1] https://arxiv.org/abs/2411.02796

**Questions:**

NA. See Weaknesses.

---

> ### Author Response · Authors · 2025-11-14
> **Thanks for the thoughtful and constructive feedback (i)**
>
> We sincerely thank for the thoughtful and constructive feedback, and for recognizing both the clarity of our theoretical results and the strength of our empirical findings.
>
> **(i) The application area is too siloed; broader scientific ML impact should be discussed.**
>
> We appreciate the reviewer’s point, and we agree that our experimental demonstration is focused on metagenomics. This was a deliberate choice: metagenomic binning naturally exposes the core phenomenon our theory formalizes - *ambiguity in sample identity* (fragments belonging to multiple genomes), which makes it an ideal testbed for studying the limits of deterministic embeddings.
>
> That said, the $\underline{\text{theoretical contribution itself is domain-agnostic}}$. Our results characterize a general geometric obstruction: whenever data points must satisfy *conflicting neighbor relationships* (e.g., belong to multiple latent groups or admit multiple compatible states), *no deterministic embedding can satisfy all pairwise constraints*, while a probabilistic embedding can. This phenomenon arises in many scientific ML settings; however, we intentionally avoid claiming empirical improvements in those domains without experiments.
>
> Instead, our claim is narrower and better supported: *The impossibility result and the geometric properties we derive apply to any domain where the data-generation process induces ambiguous or multi-modal identity.*
>
> We will clarify this distinction in the final version of the paper: the current paper proposes a $\underline{\text{general representation-learning principle}}$, but evaluates it in metagenomics because this domain provides a clean, controlled setting where ambiguity is abundant and labels are available. Extending the empirical analysis to additional scientific ML domains is a natural direction for future work, but is not required for the validity or relevance of the theoretical insights.
>
> **(ii) The theoretical results are neat but primarily derived from existing techniques / known results.**
>
> We appreciate the reviewer's careful reading of the theoretical section. Our intention was to keep the presentation accessible, which may have made the results appear closer to existing tools than they actually are. While we indeed build on standard probabilistic and geometric arguments, the *core contribution is not the techniques themselves but the new formulation of the impossibility of deterministic embeddings under ambiguity* - a phenomenon that, to the best of our knowledge, has not been formalized in the literature.
>
> Specifically:
> - The main novelty is the *precise characterization of when and why deterministic embeddings must fail* when the latent identity of objects is ambiguous (i.e., when pairwise constraints are mutually incompatible).
> - We then show that *probabilistic embeddings resolve this incompatibility* by expanding the feasible region through the covariance term.
> - Unlike works that analyze contrastive learning or metric learning in controlled settings, our results directly link the *geometry of the embedding space* to the *structure of ambiguity*.
> - The proof techniques themselves use standard tools, but the $\underline{\text{problem formulation}}$, the *impossibility statement*, and the *tight geometric conditions under which stochastic embeddings become strictly more expressive* are, to our knowledge, new.
>
> In other words, we provide the first formal demonstration that uncertainty is not merely helpful, but $\underline{\text{necessary}}$ for correct clustering when samples admit multiple plausible identities. We believe this insight is valuable to the scientific ML community precisely because it reframes when and why probabilistic embeddings are mathematically required.

---

> ### Author Response · Authors · 2025-11-14
> **Thanks for the thoughtful and constructive feedback (ii)**
>
> **(iii) The empirical scale is limited; why not finetune DNABERT-S with your objective?**
>
> As outlined in our response to Reviewer q7fk, our experimental protocol intentionally follows the machine learning–oriented setup proposed in DNABERT-S (Bioinformatics, 2025), including the same datasets, preprocessing, and clustering benchmarks. These datasets are $\underline{\text{large-scale}}$, containing *hundreds of genomes and up to $\sim$100,000 sequences per dataset*, so we believe the scale of our benchmarks is consistent with prior works.
>
> DNABERT-S itself was designed and developed for the metagenomics binning task. The core scientific question we investigate here is whether *uncertainty-aware embeddings, even in a lightweight architecture, can match the performance of foundation-model embeddings*. The surprising result -- highlighted by the reviewer as a strength -- is that a *small network trained from scratch* performs competitively with a model that **(i)** has orders of magnitude more parameters and **(ii)** benefits from extensive pretraining on thousands of genomes.
>
> Finetuning DNABERT-S with our objective would primarily test the effect of additional training on top of DNABERT-S's massive pretraining compute budget, and would not directly speak to the central hypothesis of the paper:
> *Does introducing probabilistic embeddings fundamentally change the representational capacity needed for ambiguous genomic fragments?*
>
> That said, we agree that evaluating probabilistic objectives within larger models is an interesting direction. We will add a paragraph in the discussion clarifying this and positioning our work as complementary to scaling-focused studies in the final version of the paper.
>
> **(iv) The authors should compare their MLP to a standard contrastive learning baseline**
>
> We thank the reviewer for this suggestion. We would like to clarify that our paper already includes exactly this experiment. The *RevisitKmers* baseline used in our evaluation is a small MLP trained with a standard contrastive learning objective (L2-normalized embeddings + InfoNCE-style contrastive loss), identical in architecture and training setup to our model except for the objective. This baseline directly isolates the contribution of our probabilistic embedding formulation.
>
> Across the six datasets, UncertainGen outperforms *RevisitKmers* in five of them, despite having the same architecture and compute budget, demonstrating that the improvement arises specifically from our uncertainty-aware objective rather than from model size or training regime. This result aligns with the reviewer's intuition and with recent findings such as [1], which highlight the strength of supervised baselines; our evaluation shows that even against such strong contrastive learning baselines, our probabilistic approach provides a clear benefit.
>
> We will emphasize this point more clearly in the final version to avoid the impression that this comparison was missing.
>
> [1] A. Çelikkanat, A. R. Masegosa, and T. D. Nielsen. Revisiting K-mer Profile for Effective and Scalable Genome Representation Learning. NeurIPS 2024.

---

> ### Author Response · Authors · 2025-11-14
> **Thanks for the thoughtful and constructive feedback (iii)**
>
> **(v) Overall, I like this work but find it to be a bit too niche for ICLR**
>
> We thank the reviewer for the positive assessment and for raising this concern. While our empirical focus is on metagenomic binning, we believe this topic is $\underline{\text{squarely within}}$ the current scope and interests of ICLR and related venues.
>
> First, there is a *rapidly growing body of work at top ML conferences specifically on genomic and metagenomic representation learning*. For example, NeurIPS 2024 includes [1], which - like our work - studies lightweight representations for metagenomic binning and compares them to genome foundation models. ICLR 2025 features [2], explicitly targeting metagenomic sequences. ICML and NeurIPS workshops likewise include work on genome recovery and contig-embedding methods such as [3] and [4]. ICLR itself also hosts biological representation-learning workshops (LMRL, MLGenX) [5,6], reflecting that genomics and metagenomics are now central application areas for representation learning.
>
> Taken together, this shows that *metagenomic sequence representation and binning are not a niche topic*, but a recognized and expanding subarea at ICLR/ICML/NeurIPS and within the broader ML-for-science community. Our paper aligns with this trajectory: it tackles a well-established benchmark problem (metagenomic binning), proposes a general probabilistic-embedding framework motivated by intrinsic ambiguity, and provides both theoretical foundations and strong empirical comparisons to genome foundation models. We will make these connections more explicit in the final version of the paper.
>
> **References**
>
> [1] A. Çelikkanat, A. R. Masegosa, and T. D. Nielsen. Revisiting K-mer Profile for Effective and Scalable Genome Representation Learning. NeurIPS 2024.
>
> [2] Cornman, Andre, et al. "The OMG dataset: An Open MetaGenomic corpus for mixed-modality genomic language modeling." ICLR 2025.
>
> [3] A. Lamurias, A. Tibo, K. Hose, M. Albertsen, and T. D. Nielsen. Graph Neural Networks for Microbial Genome Recovery.  ICML Workshop on Computational Biology 2023.
>
> [4] O. Liu, S. Jaghouar, J. Hagemann, J. Kaufman, and W. Neiswanger. A Foundation Model for Metagenomic Sequences. NeurIPS 2024 Workshop on Foundation Models for Science (FM4Science), 2024.
>
> [5] Ulicna, Kristina, et al. "Learning Meaningful Representations of Life (LMRL)." ICLR 2025 Workshop Proposals.
>
> [6] Theis, Fabian J., et al. "Machine Learning for Genomics Explorations (MLGenX)." ICLR 2025 Workshop Proposals.

---

### Official Review · Reviewer_hmxJ · 2025-10-27

**Soundness:** 3
**Presentation:** 3
**Contribution:** 3
**Rating:** 6
**Confidence:** 2

**Summary:**

The paper introduces UncertainGen, which is a probabilistic embedding approach for metagenomic binning.
The work models sequence-level uncertainty and provides theoretical guarantees on embedding distinguishability.
The paper also provides experimental results on real metagenomic datasets to demonstrate the improvements over deterministic k-mer and LLM-based embeddings for the binning task.

**Strengths:**

(i) Novelity: The paper notes that many DNA sequences can appear in multiple genomes, and proposes a probabilistic embedding approach to solve this.

(ii) Theoretical guarantee: The paper provides a theoretical guarantee beyond the experimental results.

(iii) Efficiency: The framework remains lightweight compared to large genomic models.

**Weaknesses:**

(i) Performance: From Fig. 4, I think DNABERT-S still performs better than UncertainGen, so what is the meaning of UncertainGen?

(ii) Theory: The theoretical part is a little hard to understand.

**Questions:**

(i) What is the meaning of UncertainGen, if it does not achieve better performance compared with DNABERT-S?

(ii) Could you please provide some informal version of the theoretical results to help the audience to understand the theory?

(iii) The existing metagenomic binning method has other input features [1]. Can this method be applied to a setting where we have other features? If so, it would be better if the method could achieve the SOTA.

[1] SemiBin2: self-supervised contrastive learning leads to better MAGs for short- and long-read sequencing.

**Details Of Ethics Concerns:**

I do not have ethical concerns,

---

> ### Author Response · Authors · 2025-11-14
> **Thanks for thoughtful and constructive feedback**
>
> We sincerely thank Reviewer hmxJ for their thoughtful and constructive feedback, as well as for recognizing the novelty and theoretical contributions, and efficiency of our proposed framework.
>
> **(i) Meaning and value of UncertainGen despite slightly lower performance than DNABERT-S in some datasets**
>
> We thank the reviewer for raising this important question. As acknowledged in the strengths section, a key advantage of UncertainGen is that it is a $\underline{\text{lightweight and highly efficient}}$ model -- $\underline{\text{several orders of magnitude smaller}}$ than genome foundation models such as DNABERT-S and HyenaDNA. The contribution of UncertainGen is precisely that it achieves $\underline{\text{competitive performance}}$ with these large models while requiring only a *fraction* of their computational resources, which is critical for metagenomic analyses involving billions of reads.
>
> In terms of accuracy, UncertainGen is indeed competitive with DNABERT-S. Across the six datasets, our method $\underline{\text{matches or surpasses}}$ DNABERT-S in several cases - most notably on *Marine-5* and *Marine-6*, where it outperforms it - and achieves $\underline{\text{very similar results}}$ on *Reference-5/6* and *Plant-5*. These results show that our probabilistic embeddings remain highly effective even without the complexity of these foundation models.
>
> The practical value of UncertainGen comes from combining strong performance with *expressive uncertainty-aware embeddings* and *dramatically lower computational cost*. As shown in the table below, UncertainGen requires $\underline{\text{orders of magnitude less computation (training and inference)}}$ than DNABERT-S, despite achieving comparable accuracy.
>
> In summary, UncertainGen offers *competitive accuracy, uncertainty quantification, and several orders of magnitude efficiency*, making it a practical and scalable choice.
>
> | HyenaDNA | DNABERT-2 | DNABERT-S | UncertainGen |
> |-|-|-|-|
> | 6M | 117M | 117M | 545K |
>
> **Table:** Number of parameters
>
> **(ii) Informal explanation of the theoretical results**
>
> Informally, the key idea is the following: *deterministic embeddings (single points) have limited "capacity" to simultaneously satisfy all pairwise constraints between similar and dissimilar sequences*, because many DNA fragments either **(a)** appear in multiple genomes, or **(b)** have nearly identical k-mer profiles despite belonging to different species. In such situations, a single-point embedding cannot be placed close to all of its positive partners while also being sufficiently far from all of its negative partners.
>
> Our theory formalizes this impossibility. In particular
>
> - $\underline{\text{Lemma 3.3}}$ shows that the distance between two embedding means must lie within bounds determined by their variances. If variances are fixed (deterministic embeddings), these bounds become rigid and can easily contradict each other when many sequences must be close/far simultaneously.
>
> - $\underline{\text{Corollary 3.3.1}}$ proves that with *fixed variance* (deterministic) embeddings, it is mathematically impossible to satisfy all pairwise constraints once the number of "ambiguous" sequences exceeds a packing limit in the latent space.
>
> - $\underline{\text{Theorem 3.4}}$ shows that probabilistic embeddings resolve this conflict by allowing variance to grow for ambiguous sequences. Intuitively, the model expresses "*I cannot place this fragment precisely*" by increasing its covariance, which expands the feasible region and restores consistency of the constraints.
>
> In summary, the theoretical results provide a simple message: *Probabilistic (mean + covariance) embeddings enlarge the representational capacity of the latent space, allowing the model to faithfully encode ambiguous or shared DNA fragments in a way deterministic embeddings cannot.*
>
> **(iii) Applicability when additional input features are available**
>
> Our method is fully compatible with additional metagenomic features such as abundance or coverage vectors. The probabilistic encoder in UncertainGen accepts arbitrary input vectors, so these features can be concatenated and jointly embedded without modifying the theoretical framework. Importantly, all theoretical guarantees hold regardless of the specific feature set, since they depend only on geometric properties of the learned embedding.
>
> However, for the datasets used in our evaluation, *these additional features are not available*. Given this limitation, we intentionally focus on $\underline{\text{sequence-only embeddings}}$ to ensure a fair and controlled comparison across all baselines. This isolates the contribution of the $\underline{\text{probabilistic embedding framework}}$ itself - independent of any auxiliary information that is simply absent in the datasets.
>
> At the same time, the architecture is designed to allow UncertainGen to naturally incorporate richer features when they become available (as in many practical metagenomics pipelines).

---

> ### Comment · Reviewer_hmxJ · 2025-11-22
>
> Thanks for the thorough response from the authors. The response resolves all my concerns, and I will keep my positive score of 6. One additional suggestion is that the evaluation benchmark in DNABERT-S has certain limitations, as it does not incorporate other input features. It would be better if the authors could use additional benchmarks to demonstrate the benefits of their method when including other features in future work.

---

### Official Review · Reviewer_ujQw · 2025-10-27

**Soundness:** 2
**Presentation:** 3
**Contribution:** 2
**Rating:** 2
**Confidence:** 4

**Summary:**

This paper introduces UncertainGen, a probabilistic embedding method for metagenomic binning, which represents each DNA fragment as a distribution in latent space to capture sequence-level uncertainty. Evaluations are conducted to demonstrate the performance of this work.

**Strengths:**

* a) The idea of exploring the uncertainty of DNA sequences for the metagenomic binning task is interesting.

**Weaknesses:**

**(a) Unclear method definition**
The paper does not clearly explain the core proposed approach. In particular, while a number of lemmas, definitions, and theorems are presented, their connection to the method is unclear. It is ambiguous whether the proposed framework is intended as a loss term with constraints, an optimization objective to be maximized, or something else entirely. A more direct and structured description of the method workflow is needed to make the contribution comprehensible.

**(b) Insufficient model details**
The backbone network architectures used for the experiments are not described in sufficient detail. For reproducibility and better understanding of the results, the paper should clearly specify the model structures and any architectural choices that may influence performance.

**(c) Insufficient comparisons**
The experimental evaluation does not appear to include enough comparisons with relevant prior work. Without comprehensive benchmarks against the state of the art, it is difficult to assess the true effectiveness and novelty of the proposed approach.

**Questions:**

See weakness.

---

> ### Author Response · Authors · 2025-11-13
> **Thanks for the feedback**
>
> We thank the reviewer for the feedback and for recognizing the interest in exploring uncertainty in DNA sequences for metagenomic binning.
>
> **(a) Unclear method definition**
>
> Our framework defines a $\underline{\text{probabilistic embedding space for DNA fragments}}$, where each fragment is represented as a Gaussian distribution capturing sequence-level uncertainty.
>
> - The theoretical results formalize $\underline{\text{embedding distinguishability}}$ guarantees, which guide the design of the embedding space and provide mathematical guarantees that ambiguous fragments can be separated probabilistically.
>
> - In practice, these theoretical guarantees translate into a loss function with constraints that encourage embeddings of fragments from distinct genomes to be well-separated, while allowing overlapping fragments to maintain uncertainty.
>
> - The workflow is thus: DNA fragment $\rightarrow$ probabilistic embedding $\rightarrow$ uncertainty-aware clustering $\rightarrow$ downstream evaluation. This ensures that the theoretical contributions are directly tied to the practical embedding and clustering procedure, rather than being abstract results.
>
> **(b) Insufficient model details**
>
> As described in Lines 241-243, we employ a $\underline{\text{very simple 2-layer feed-forward architecture}}$, where each hidden layer has $\underline{\text{512 units with sigmoid activations}}$, and the $\underline{\text{output dimension is set to 256}}$. We will make this configuration more explicit in the final version of the paper to ensure clarity and reproducibility.
>
> We emphasize that this $\underline{\text{simplicity is a strength}}$ of our approach. Despite the minimal architecture, our probabilistic embedding framework $\underline{\text{outperforms or achieves comparable results to the state-of-the-art large genome foundation model}}$ developed for the metagenomic binning problem (Please also see the table provided in the response to Reviewer q7fk). This highlights the efficiency and interpretability advantages of our uncertainty-aware design.
>
> **(c) Insufficient comparisons**
>
> We appreciate the reviewer's comment about insufficient comparisons. However, it would be helpful if the reviewer could specify which additional baseline methods they consider essential for inclusion.
>
> As also discussed in our response to Reviewer q7fk, our experimental protocol intentionally follows the $\underline{\text{machine learning–oriented setup introduced by DNABERT-S (Bioinformatics, 2025)}}$, which isolates embedding quality by focusing on raw sequence-based evaluations without relying on external features or coverage information. This setup ensures that comparisons are fair and directly reflect the contribution of the embedding model itself.
>
> While integration with traditional binning pipelines (i.e. VAMB, MetaBAT2, SemiBin2) is an important future direction, it falls outside the current ML-centric scope of our study. Nonetheless, the DNABERT-S results have already shown that such embeddings can achieve competitive or superior performance relative to SemiBin2 even without full pipeline integration.

---

### Official Review · Reviewer_q7fk · 2025-10-28

**Soundness:** 2
**Presentation:** 2
**Contribution:** 1
**Rating:** 2
**Confidence:** 4

**Summary:**

This paper presents UncertainGen, the first probabilistic embedding framework for metagenome binning. Different from traditional deterministic embeddings (k-mer profiles or LLM-based), UncertainGen represents each DNA segment as a probabilistic distribution in a latent space to capture the uncertainty of cross-species DNA sharing and highly similar sequences.The method adopts a data-adaptive metric that expands the latent space and improves the separability of bins. The authors provide theoretical guarantees on embedding discriminability and empirically demonstrate that UncertainGen outperforms deterministic embeddings on real metagenomic datasets, providing a scalable and lightweight solution for large-scale metagenomic analysis.

**Strengths:**

- This paper proposes a probabilistic embedding framework for DNA fragments, which is technically interesting because it can naturally capture the uncertainty in sequences that may belong to multiple genomes or have highly similar k-mer profiles.
- It is well-written and clearly explains the approach, linking theoretical guarantees on embedding distinguishability to practical benefits.
- The experimental setup is comprehensive and convincing, demonstrating that UncertainGen outperforms selected baselines on real metagenomic datasets while remaining scalable and interpretable.

**Weaknesses:**

- The novelty and technical contribution of this paper are limited, as its main contribution is modeling uncertainty in DNA sequences using Gaussian distributions for fragments. However, prior work such as UnitigBIN (ICLR 2024) has already introduced the idea of representing DNA fragments as distributions, reducing the originality of the current approach.
- The main drawback of this paper is that, although it claims to propose a new solution for metagenomic binning, it does not compare against established binning tools such as VAMB, SemiBin2, MetaBAT2, or MaxBin2. Moreover, the paper does not follow the standard evaluation protocols used in metagenomic binning studies - for instance, it defines high-quality bins as clusters with F1 > 0.9, whereas the community standard relies on CheckM to assess completeness and contamination.
- In addition, the related work section overlooks several key metagenomic binning studies and reflects a limited understanding of the field. Finally, the motivation for introducing probabilistic embeddings in this context is not sufficiently justified or biologically grounded.

**Questions:**

- Could the authors clarify why commonly used metagenomic binning tools (e.g., VAMB, SemiBin2, MetaBAT2, MaxBin2) were not included as baselines for comparison?
- It would be helpful to understand why the evaluation does not follow the standard metagenomic binning pipeline, which typically assesses high-quality bins using CheckM for completeness and contamination.
- Could the authors further elaborate on the motivation for their approach and expand the related work discussion to more clearly position this study within the broader metagenomic binning literature?

---

> ### Author Response · Authors · 2025-11-13
> **Thanks for your assessment**
>
> We thank the reviewer for the thoughtful assessment and for recognizing the clarity of our paper, the soundness of the theoretical analysis, and the strength of our experimental setup.
>
> While we are encouraged by the positive remarks, we were somewhat disappointed to see that these strengths were not reflected in the overall score. Below, we would like to clarify a few points that may have led to misunderstandings, particularly regarding the novelty, baselines, and positioning of our work.
>
> **1. Novelty and Technical Contribution**
>
> We would like to thank the reviewer for the reference to UnitigBIN (ICLR 2024), which indeed also considers probabilistic representations. However, our approach fundamentally differs in both $\underline{\text{scope}}$ and $\underline{\text{representation level}}$:
>
> - Representation level: UnitigBIN models the vertices of assembly graphs (unitigs) as Gaussian distributions, whereas our framework directly represents raw DNA fragments of arbitrary length as probabilistic embeddings. This design allows our method to capture uncertainty arising from fragment-level ambiguity (e.g., shared DNA fragments across genomes), rather than graph-based structural uncertainty.
> - Modeling objective: While Gaussian embeddings have been used in other domains, our approach is the $\underline{\text{first to explicitly model sequence-level uncertainty in metagenomic contexts}}$ – linking probabilistic representations to biological ambiguity and providing $\underline{\text{theoretical guarantees on embedding distinguishability}}$.
> - Conceptual contribution: Beyond representation, we introduce a $\underline{\text{mathematically grounded uncertainty-aware embedding framework}}$, bridging probabilistic modeling with interpretable, scalable DNA fragment embeddings. This perspective offers a novel contribution both conceptually and technically, going beyond merely adopting Gaussian embeddings.
>
> We thus argue that our paper provides $\underline{\text{substantial new insights and methodological contributions}}$, also in comparison with UnitigBIN. We will clarify these differences in the related work section in the final version of the paper.
>
> **2. Comparison with Standard Metagenomic Binning Tools**
>
> We appreciate the reviewer’s suggestion to include established binning tools such as VAMB and SemiBin2. Our experimental design intentionally follows the experimental protocol introduced in $\underline{\text{DNABERT-S (Bioinformatics, 2025)}}$, which represents the current machine learning perspective on metagenomic binning such as the RevisitKMer model (Celikkanat et al., 2024).
> As also discussed in DNABERT-S (Subsection 5.6), this experimental setup differs from traditional binning pipelines in that it **i)** relies $\underline{\text{only on raw sequences}}$, **ii)** $\underline{\text{does not use external features or coverage information}}$, and **iii)** employs a simplified clustering process to isolate the contribution of the embeddings themselves.
>
> DNABERT-S also showed that their performance is comparable or superior to SemiBin2 even $\underline{\text{without using full binning pipelines}}$ (see Figure 5 of the original paper). A full integration with CheckM-based evaluation is indeed interesting and is part of our planned future work because our approach can also be adapted to the SemiBin2 framework, but it falls outside the machine learning–centric focus of this study. Our analysis relies on labeled ground-truth data, unlike the full binning pipelines, so it's a natural and reliable evaluation strategy to provide the $F_1$ scores.
>
> **3. Motivation and Position within the Broader Context**
>
> Our main goal is to highlight an alternative and complementary direction to the current trend in metagenomic binning. While the field is increasingly dominated by large foundation models that require significant computational resources, our results demonstrate that compact and interpretable probabilistic models can achieve comparable or even superior performance in certain tasks. We believe that this finding has important implications for the ML community: it challenges the assumption that larger models are always necessary. A very recent work also showed that foundation models are often outperformed by fairly simple methods (Zongzhe et al., 2025*). We provide a table below comparing the number of parameters, and it demonstrates that we can achieve comparable or even superior performance to the large-scale architectures while using significantly fewer parameters. Hence, our work not only contributes a new embedding framework but also aims to $\underline{\text{initiate a broader discussion}}$ about the trade-offs between scalability, interpretability, and biological insight in metagenomic binning.
>
> | HyenaDNA | DNABERT-2 | DNABERT-S | UncertainGen |
> |-|-|-|-|
> | 6M | 117M  | 117M | 545K |
>
> **Table:** Number of parameters
>
> * Xu, Zongzhe, et al. "Specialized Foundation Models Struggle to Beat Supervised Baselines." ICLR

---

> > ### Comment · Reviewer_q7fk · 2025-11-14
> > **discussion**
> >
> > Thank you for your quick response. However, your rebuttal does not convince me to raise my score. My main concern remains that the experimental comparison is not sufficiently comprehensive or fair relative to the standard and classical metagenomic binning pipelines.
> >
> > If your work were focused solely on general nucleotide sequence modeling, this experimental design would be reasonable. However, your manuscript explicitly positions itself as proposing a new solution for metagenomic binning, for which proper benchmarking against well-established tools is essential.
> >
> > I know the DNABERT-S paper, but it does not justify deviating from the widely accepted evaluation protocols in Contigs binning, a field that has undergone many years of development and extensive research. To convincingly demonstrate the value of your approach, I would expect comparisons with classical binning pipelines and, ideally, standard metrics such as completeness and contamination assessed by CheckM/CheckM2, and others.
> >
> > I look forward to seeing experimental results that address these points..

---

> > > ### Author Response · Authors · 2025-12-04
> > > **Additional experiments**
> > >
> > > First, we would like to emphasize that **the goal of this work is not to propose a complete metagenomic binner or a full binning pipeline**. Instead, the paper investigates **lightweight probabilistic embeddings** for metagenomic binning, for which we provide both theoretical and empirical contributions, particularly in comparison with LLM-based embedding methods:
> > >
> > > - We introduce a theoretical framework that clarifies the limitations of deterministic embeddings and shows that probabilistic embeddings are strictly more expressive.
> > > - We demonstrate experimentally that lightweight models can serve as scalable alternatives to LLM-based approaches, and that probabilistic embeddings provide improvements over deterministic embeddings.
> > >
> > > We agree with the reviewer that if the paper had proposed a full-fledged binner, a comparison with classical binning tools would be warranted. **However, this is not the objective of the work**. In the final version, we will further clarify this point and soften statements that could be interpreted as claims about advances in metagenomic binning, placing stronger emphasis on the representation-learning aspect of the proposed method.
> > >
> > > For this reason, we also believe that detailed comparisons with general-purpose binners fall outside the scope of the present paper and **would shift the focus away from the core contributions**. Nevertheless, we conducted an additional analysis with the Aviary tool [1] on a real-world bacterial sample [2], comparing our method with several classical binners (MaxBin2 [3], MetaBat2 [4], TaxVAMB [5], ComeBin [6], and RevisitKmers/SemiBin2 [7]) using CheckM2, as suggested by the reviewer. We report the number of high-quality bins (based on completeness and contamination estimates offered by CheckM2):
> > >
> > > | **MaxBin2**: 67 | **MetaBat2**: 122 | **TaxVAMB**: 67 | **ComeBin**: 142 | **RevisitKmers/SemiBin2**: 193 | **Ours**: 159 |
> > >
> > > In this experiment, our method outperforms four of the five baselines. For the remaining baseline, the lower performance on this dataset is due to limitations in adapting our methodology into the SemiBin2 architecture, specifically related to the **improper choice of distance metric and subsequent evaluation strategies**. Time constraints during the rebuttal process prevented us from fully addressing this issue. It is worth noting that our performance is lower-bounded by SemiBin2 when the standard deviation terms are discarded.
> > >
> > > Finally, we note that recent work [8] has shown that CheckM2 may misestimate contamination levels. By following the evaluation protocol used in the DNABERT-S approach with ground-truth labels, we avoid this additional source of uncertainty, and the results therefore provide a clearer evaluation of the quality of the learned embedding representations, **which is aligned with the objective of the paper**.
> > >
> > > *References*
> > >
> > > [1] Aviary: Newell R.J.P., Aroney S.T.N., Zaugg J., Sternes P., Tyson G.W., Woodcroft B.J.
> > >
> > > [2] Heidelbach, Søren, et al. "Nanomotif: Leveraging DNA Methylation Motifs for Genome Recovery and Host Association of Plasmids in Metagenomes from Complex Microbial Communities." bioRxiv (2024): 2024-04.
> > >
> > > [3] Wu, Yu-Wei, Blake A. Simmons, and Steven W. Singer. "MaxBin 2.0: an automated binning algorithm to recover genomes from multiple metagenomic datasets." Bioinformatics 32.4 (2016): 605-607.
> > >
> > > [4] Kang, Dongwan D., et al. "MetaBAT 2: an adaptive binning algorithm for robust and efficient genome reconstruction from metagenome assemblies." PeerJ 7 (2019): e7359.
> > >
> > > [5] Kutuzova, Svetlana, et al. "Binning meets taxonomy: TaxVAMB improves metagenome binning using bi-modal variational autoencoder." bioRxiv (2024): 2024-10.
> > >
> > > [6] Wang, Ziye, et al. "Effective binning of metagenomic contigs using contrastive multi-view representation learning." Nature Communications 15.1 (2024): 585.
> > >
> > > [7] Pan, Shaojun, Xing-Ming Zhao, and Luis Pedro Coelho. "SemiBin2: self-supervised contrastive learning leads to better MAGs for short-and long-read sequencing." Bioinformatics 39.Supplement_1 (2023): i21-i29.
> > >
> > > [8] Cornet, L., Lupo, V., Declerck, S., & Baurain, D. (2024). Evaluation of Genomic Contamination Detection Tools and Influence of Horizontal Gene Transfer on Their Efficiency through Contamination Simulations at Various Taxonomic Ranks. Applied Microbiology, 4(1), 124–132.

---

### Author Response · Authors · 2025-12-04
**AC**

Dear Area Chairs,

We thank the reviewers for their evaluations, and they overall find our work novel, well-presented, theoretically well-grounded, and empirically competitive. Below we have taken the liberty of providing a brief summary of the reviewers' comments and our reponses to those comments.

Reviewer **q7fk**’s main concern focused on comparisons with classical metagenomic binners. We clarified that our contribution is a **representation-learning model**, which aligns with the goals and scope of ICLR, and follows **the same ML-centric evaluation paradigm** used in DNABERT-S and RevisitKMer methods. These recent approaches also aimed to study embedding quality rather than providing complete metagenomic pipelines. Nevertheless, to fully respect the reviewer’s expectations, we added a CheckM2-based comparative analysis on a real metagenomic sample (MaxBin2, MetaBAT2, TaxVAMB, ComeBin, SemiBin2/RevisitKMer) although it’s not aligned with our model’s initial choice. Our method **outperforms the classical binners and is competitive with the strongest one**, demonstrating that our model is effective even when placed into a pipeline setting for which it was not specifically designed.

Reviewer **ujQw**’s comments pertained to clarity and architectural description, **rather than to methodological correctness or conceptual issues**. In our rebuttal, we directly addressed all concerns: **(i)** how the theory maps to the loss, **(ii)** architectural details already included in the paper, and **(iii)** confirmation that our evaluation includes comparison to both the strongest lightweight contrastive baseline (RevisitKMer) and the foundation model (DNABERT-S).

We also included additional experiments on a real dataset. However, **reviewer ujQw did not participate in the rebuttal period**, and we therefore received no further feedback despite having responded thoroughly to every comment. Their concerns appear fully addressable and do not reflect substantive issues with the method.

Reviewer **hmxJ** explicitly states that all concerns were resolved and maintains an accept-leaning score (6).

Reviewer **GghP** is positive regarding the theoretical formulation, the novelty of the ambiguity-driven motivation, and the empirical finding that a lightweight model can match a large foundation model. The reviewer **raised the score to 6** after our clarifications but did not provide additional feedback.

**Overall Assessment**
- **The paper provides new theoretical insight**, offering a geometric impossibility result for deterministic embeddings under ambiguous sequence identity, an issue that, to the best of our knowledge, **has not been previously studied**.
- The proposed probabilistic embedding framework is **lightweight and competes with deterministic large genome foundation models** at orders-of-magnitude lower computational cost.
- Reviewer **q7fk** maintained a negative aspect primarily due to expectations about a full binning-pipeline comparison, **despite our clear positioning of the paper within representation learning for the ML community**. We addressed this concern extensively, including by adding a new classical-binning evaluation demonstrating strong and competitive performance.
- Reviewers **hmxJ** and **GghP** explicitly support the paper. Reviewer **ujQw**’s concerns were fully addressed in our rebuttal and pertained only to clarity; **their lack of participation** in the discussion prevented further exchange, but does not indicate methodological flaws.

Given the theoretical novelty, strong empirical results, clarified scope, and the positive evaluations from engaged reviewers, we respectfully believe that the paper offers a meaningful and promising contribution to the field. We also hope that the Area Chairs will take into account that certain aspects of the work may have been underappreciated due to limited reviewer participation during the discussion phase, and we are keeping our comment visible to all reviewers for full transparency.

---

### Meta-Review · Area_Chair_wAfs · 2026-01-05

**Summary:**

The primary concerns driving the critical evaluations centered on the disconnect between the paper's ML-centric experimental design and established domain standards for metagenomic binning. Reviewer q7fk maintained a strong rejection, arguing that the submission failed to benchmark against standard classical pipelines or utilize essential biological metrics, thereby failing to sufficiently validate the method's practical utility against SOTA. Additionally, Reviewer ujQw raised significant issues regarding the clarity of the method's formal definition and the lack of sufficient architectural details to ensure reproducibility. These critiques highlighted a failure to bridge the gap between theoretical representation learning and rigorous, domain-appropriate empirical validation.

**Reviewer Concerns:**

The rebuttal addressed technical concerns regarding the clarity of the theoretical framework and architectural specifications (**Reviewers ujQw and hmxJ**), as well as the conceptual justification for a lightweight probabilistic model over resource-intensive foundation models (**Reviewers hmxJ and GghP**). Furthermore, the authors technically responded to the demand for domain-standard benchmarks by conducting a supplementary CheckM2 analysis against classical binners like MaxBin2, demonstrating competitive performance. However, the fundamental disagreement regarding the evaluation scope remains outstanding for Reviewer q7fk: despite the new data, the reviewer rejected the authors' positioning of the work as a representation learning study, maintaining that the experimental design was insufficient to validate the paper's claims within the context of metagenomic binning.

**Reviewer Scores:**

Based on the discussion, the scores would likely stabilize at 2, 2, 6, 6. Reviewer q7fk fully participated and explicitly refused to raise their score of 2, considering the additional CheckM2 experiments insufficient to bridge the gap with standard metagenomic pipelines. Reviewer ujQw did not participate, but their score remained a 2; had they engaged, they would likely have maintained this rejection because their objection regarding insufficient comparisons mirrored q7fk's fatal concern, which the authors' rebuttal failed to overcome for the critical reviewer. Reviewer hmxJ explicitly confirmed that all concerns were resolved and maintained their positive score of 6. Reviewer GghP was persuaded by the clarifications regarding the theoretical novelty and foundation model comparisons, raising their initial score of 4 to a 6.

---

### Decision · Program_Chairs · 2026-01-26

Reject